# Probabilistic embedding, clustering, and alignment for integrating spatial transcriptomics data with PRECAST

Wei Liu [1,10], Xu Liao [1,10], Ziye Luo [1,2], Yi Yang[1], Mai Chan Lau[3], Yuling Jiao [4], Xingjie Shi [5], Weiwei Zhai [6], Hongkai Ji [7], Joe Yeong [3,8] & Jin Liu [1,9] ✉

Spatially resolved transcriptomics involves a set of emerging technologies that enable the transcriptomic profiling of tissues with the physical location of expressions. Although a variety of methods have been developed for data integration, most of them are for single-cell RNA-seq datasets without consideration of spatial information. Thus, methods that can integrate spatial transcriptomics data from multiple tissue slides, possibly from multiple individuals, are needed. Here, we present PRECAST, a data integration method for multiple spatial transcriptomics datasets with complex batch effects and/or biological effects between slides. PRECAST unifies spatial factor analysis simultaneously with spatial clustering and embedding alignment, while requiring only partially shared cell/domain clusters across datasets. Using both simulated and four real datasets, we show improved cell/domain detection with outstanding visualization, and the estimated aligned embeddings and cell/domain labels facilitate many downstream analyses. We demonstrate that PRECAST is computationally scalable and applicable to spatial transcriptomics datasets from different platforms.

Spatially resolved transcriptomics (SRT) encompass a set of recently developed technologies that characterize the gene expression profiles of tissues while retaining information on their physical location. The methodologies used for resolving spatial gene expression are primarily categorized into in situ hybridization (ISH) technologies (e.g., MERFISH[1–3], seqFISH[4,5], seqFISH+[6]), and in situ capturing technologies (e.g., ST[7], HDST[8], Slide-seq[9,10], and 10x Genomics Visium[11])[12]. The in situ capturing technologies are unbiased and involve transcriptome-wide expression measurements, while ISH-based methods are targeted and require prior knowledge of the genes of interest. These technologies have provided extraordinary new opportunities for researchers to characterize the transcriptomic landscape within a spatial context; explore how cells influence and are influenced by the cells around them[13]; identify genes with spatial variations other than cell/domain differences, e.g., cell morphology[14]; and identify spatial trajectories or RNA velocity in tissues[15,16], among other applications[17].

Similar to single-cell RNA-sequencing (scRNA-seq) studies, in SRT studies of a single slide, identifying the cell/domain clusters for each spot with the collation of both spatial information and expression measurements is an important step[18–20]. Recently, multiple studies have involved the analysis of SRT datasets from multiple slides, requiring to further remove unwanted variations from different

[1]Centre for Quantitative Medicine, Health Services & Systems Research, Duke-NUS Medical School, Singapore, Singapore. [2]School of Statistics, Renmin University, Beijing, China. [3]Institute of Molecular and Cell Biology (IMCB), Agency of Science, Technology and Research (A*STAR), Singapore, Singapore. [4]School of Mathematics and Statistics, Wuhan University, Wuhan, China. [5]Academy of Statistics and Interdisciplinary Sciences, East China Normal University, Shanghai, China. [6]Key Laboratory of Zoological Systematics and Evolution, Institute of Zoology, Chinese Academy of Sciences, Beijing, China. [7]Department of Biostatistics, Johns Hopkins Bloomberg School of Public Health, Baltimore, MD, USA. [8]Department of Anatomical Pathology, Singapore General Hospital, Singapore, Singapore. [9]School of Data Science, The Chinese University of Hong Kong-Shenzhen, Shenzhen, China. [10]These authors contributed equally: Wei Liu, Xu Liao. ✉e-mail: jin.liu@duke-nus.edu.sg

batches. For example, SRT profiles were characterized in 12 human cortex tissue slides from three adult donors using 10x Visium[21] and in multiple sections from a mouse olfactory bulb (OB) that were equally distributed along the anterior-posterior axis of the same mouse using Slide-seqV2[22]. Moreover, when multiple SRT datasets from different clinical/biological conditions are available, integrative analysis to estimate shared embeddings of expressions representing variations between cell/domain types can provide the first step towards detecting genes that are differentially expressed between conditions[23]. Thus, it is important to develop rigorous methods that are capable of performing data integration across multiple SRT datasets by aligning shared embeddings of biological effects between cell/domain types while accounting for complex batch effects and/or biological effects between slides[24].

An ideal data integration method for SRT datasets should be capable of the following three tasks: (1) estimating the shared embeddings of biological effects between cell/domain types across SRT datasets and slide-specific embeddings that account for local microenvironments (spatial dimension reduction); (2) aligning the shared embeddings that capture cellular biological variation across datasets with heterogeneous batch effects and/or biological effects between slides (data alignment); (3) clustering the aligned embeddings to obtain cell/domain clusters across datasets that promote spatial smoothness (spatial clustering). Most existing data integration methods, including MNN[24], Scanorama[25], Seurat V3[26], Harmony[23], scVI[27], and scGen[28], have been developed for scRNA-seq datasets without any consideration of spatial information. More recently, MEFISTO was proposed as a way to analyze datasets with repeated spatio-temporal measurements[29], and PASTE was proposed as a method to stack and/or integrate SRT data from multiple adjacent tissue slices into a single slide, but it is not applicable to the integration of tissue sections from different individuals[30]. In addition, most existing methods perform data integration in low-dimensional space using the principal components (PCs) of conventional dimension reduction, e.g., principal component analysis (PCA), without considering the consistent loss functions of dimension reduction, alignment across datasets, or spatial clustering.

To address the challenges presented by SRT data integration and facilitate the downstream analyses of combinations of multiple tissue slides, we propose the use of a unified and principled probabilistic model, PRECAST, to simultaneously estimate low-dimensional embeddings for biological effects between cell/domain types, perform spatial clustering, and most importantly, align embeddings for normalized gene expression matrices from multiple tissue slides. As a result, PRECAST can resolve aligned representations, provide outstanding visualizations, and achieve higher spatial clustering accuracy for combined tissue slides. The resolved aligned representations and estimated labels from PRECAST can be used in multiple downstream analyses, e.g., removing batch effects, identifying differentially expressed genes under different conditions/stages, and recovering spatial trajectories/RNA velocity, etc. In addition, PRECAST uniquely estimates slide-specific embeddings that capture the spatial dependence in neighboring cells/spots, providing an opportunity to understand the spatial impact of various microenvironments. We illustrate the benefits of using PRECAST through extensive simulations and analysis of a diverse range of example datasets collated with different spatial transcriptomics technologies: 10x Visium datasets of 12 human dorsolateral prefrontal cortex (DLPFC) samples and four hepatocellular carcinoma (HCC) samples, ST datasets from eight mouse liver tissue sections and Slide-seqV2 datasets from 16 OB tissue slides.

## Results

### Spatial transcriptomics data integration using PRECAST
Unlike other integration methods that take as input the top (spatial) PCs and apply multiple steps to remove batch effects and merge the

data, PRECAST takes as input the normalized gene expression matrices from multiple tissue slides, factorizes the input to each matrix into a latent factor with a shared distribution in each cell/domain cluster while simultaneously performing spatial dimension reduction and spatial clustering, and aligning and estimating joint embeddings for biological effects between cell/domain types across multiple tissue slides (Fig. 1a). In the dimension-reduction step, we used an intrinsic conditional autoregressive (CAR) component to capture the spatial dependence induced by neighboring microenvironments while in the spatial-clustering step, we used a Potts model to promote spatial smoothness within spot neighborhoods in the space of cluster labels. PRECAST applies a simple projection strategy to non-cellular biological effects, e.g., batch effects and/or biological effects between slides, and implicitly accounts for shifts in the centroid of each cluster using the intrinsic CAR component in the dimension-reduction step (see "Methods"). These considerations of PRECAST mimic some of the recent explorations into self-supervised learning and domain adaptation in deep learning but in a parametric manner. We show that PRECAST outperforms existing data integration methods by more successfully aligning similar clusters across multiple tissue slides while separating clusters with outstanding visualization. Uniquely, PRECAST estimates slide-specific embeddings for spatial dependence among neighboring cells/spots, providing an opportunity to explore the impact of neighboring microenvironments.

PRECAST can be applied to SRT datasets of various resolutions. By increasing the resolution for SRT datasets, PRECAST can reveal fine-scale cell-type distributions. In the following analysis, datasets in Slide-seqV2 with near-single-cell resolution present spatial patterns with much more "noise" than those of Visium due to the heterogeneity of the cell-type distributions. When we merged nearby beads in the Slide-seqV2 datasets, the recovered spatial patterns resembled those from Visium.

We use the estimated cell/domain labels and finely aligned embeddings to showcase some downstream analyses, as depicted in Fig. 1b. First, users can visualize the inferred embeddings for biological effects between cell/domain types using two components from either tSNE[31] or UMAP[32]. Second, because the aligned embeddings obtained from PRECAST only carry information on the biological differences between cell/domain types, we provide a module to recover the gene expression matrices with batch effects removed and further identify genes differentially expressed either across different cell/domain clusters and/or under different conditions. Third, using these embeddings, we can identify genes whose spatial variability is not just due to biological differences between cell/domain types. Fourth, with the aligned embeddings estimated by PRECAST, we can perform trajectory inference/RNA velocity analysis to determine either the pseudotime or the pattern of dynamics in spatial spots across multiple tissue slides.

### Validation using simulated data
We performed comprehensive simulations to evaluate the performance of PRECAST and compare it with that of several other methods (Fig. 1c and Supplementary Figs. S1–S2). Specifically, we considered the following eight integration methods: Harmony[23], Seurat V3[26], fastMNN[24], scGen[28], Scanorama[25], scVI[27], MEFISTO[29], and PASTE[30], all of which can be used to estimate the aligned embeddings among samples, except PASTE, which can only estimate the embeddings of the center slice. The simulation details are provided in the "Methods" section. Briefly, we simulated either the normalized or count matrices of gene expression to compare PRECAST with other methods in terms of performance in data integration, dimension reduction, and spatial clustering. To mimic real data, we also considered the two ways to generate spatial coordinates and cell/domain labels, i.e., Potts models and real data in three DLPFC Visium slices. Using real data in DLPFC, we considered slides either from different or the same donor. In total, we

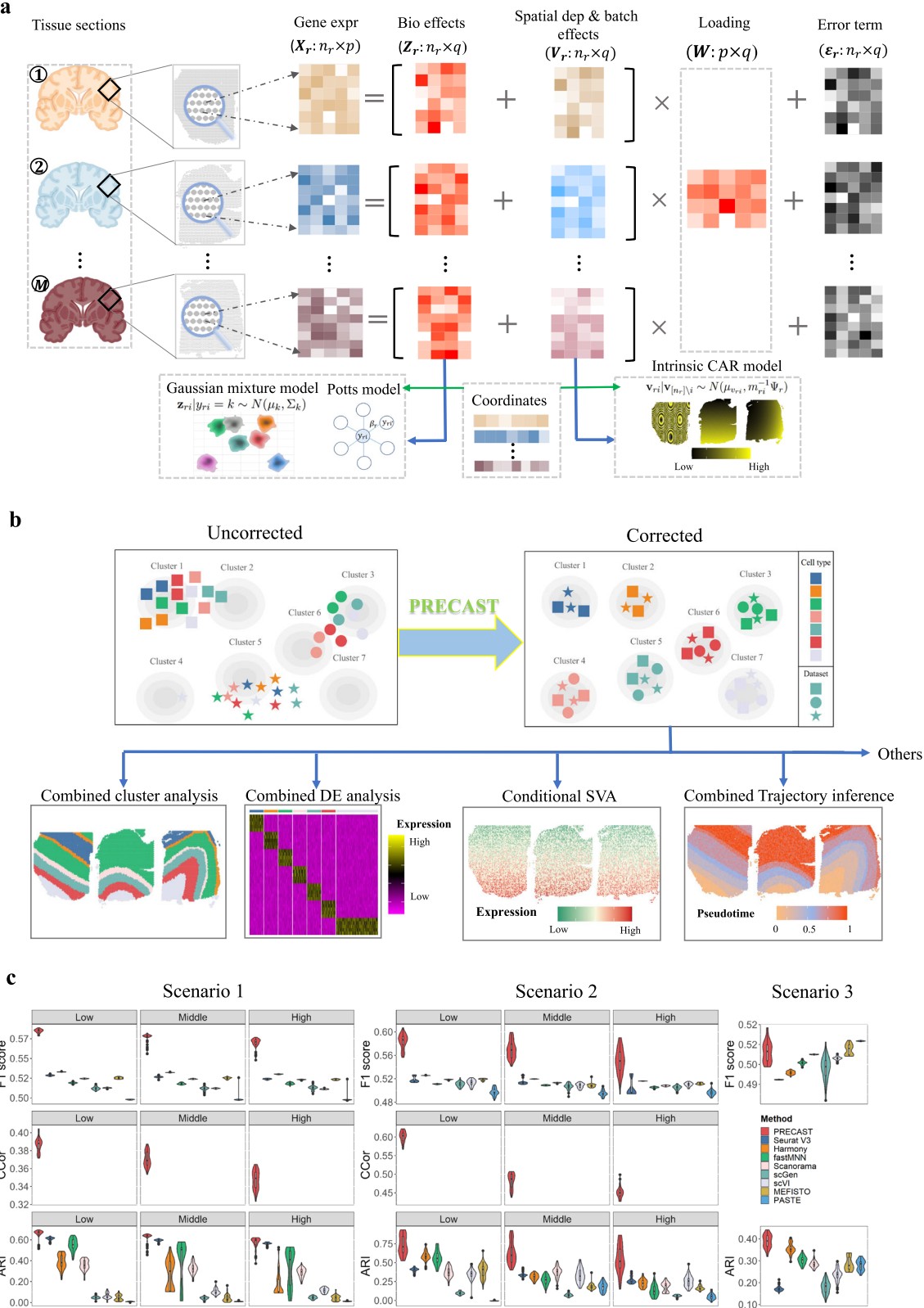

investigated five scenarios: (1) Potts + Count, with three different scales in batch effects (low, middle, high); (2) DLPFC (slides from different donors) + Count, with three different scales in batch effects (low, middle, high); (3) DLPFC (slides from the same donor) + Count; (4) Potts + logCount; and (5) DLPFC (slides from different donors) + log-Count. Scenarios 1 and 2 were used to examine the impact of scales in batch effects on data integration performance.

To quantify the performance of the data integration, we calculated the F1 scores of the average silhouette coefficients, which summarized two similar metrics of silhouette coefficients into a single quantity, and two versions of the local inverse Simpson's Index (LISI): integration LISI (iLISI) and cell-type LISI (cLISI). iLISI was employed to assess integration mixing, while cLISI was employed to assess the separation of each domain cluster. In scenarios 1 and 2, data

**Fig. 1 | Schematic overview of PRECAST and simulation results. a** PRECAST is a unified probabilistic factor model that simultaneously estimates aligned embeddings and cluster labels with consideration of spatial smoothness in both the cluster label and low-dimensional embedding spaces. Normalized gene expression matrices from multiple tissue slides are used as input. **b** Representative PRECAST downstream analyses. **c** In the simulations, we investigated two ways to generate spatial coordinates and cell/domain labels for count matrices: Potts models (scenario 1) and three cortex tissues from the DLPFC data (scenario 2). We examined the impact of scales in batch effects (low, middle, and high) on the data integration performance using scenarios 1 and 2. We also considered an additional scenario 3, which was favorable for PASTE. We evaluated performance in terms of data integration, the estimation of aligned embeddings, the estimation of slide-specific

embeddings due to neighboring microenvironments, and spatial clustering ($n = 11,425$ spots over 50 independent replicates), using F1 scores of average silhouette coefficients (F1 score), canonical correlation coefficients (CCor), and the adjusted Rand index (ARI). ARIs displayed for the other methods were evaluated based on the results of the spatial clustering method SC-MEB. PRECAST outperformed all other data integration methods in scenarios 1 and 2, and its performance was comparable to PASTE in scenario 3. In simulations, only PRECAST estimated the slide-specific embeddings. We also evaluated the CCor with underlying truth in scenarios 1 and 2. In the boxplot, the center line, box lines, and whiskers represent the median, upper, and lower quartiles, and 1.5 times interquartile range, respectively.

integration performance by all methods dropped as the scale of batch effects increased. However, PRECAST achieved the best performance in all methods, with the highest F1 scores for the average silhouette coefficients. In scenario 3, PRECAST was comparable to PASTE (Fig. 1c, top panel), and in scenarios 4 and 5, PRECAST markedly outperformed other methods (Supplementary Fig. S1c). Moreover, PRECAST was able to merge spots within common cell/domain clusters across datasets, while separating spots from different cell/domain clusters (small cLISI, Supplementary Fig. S1a, top panel; Supplementary Fig. S1c, middle panel) and, at the same time, maintaining a sufficient mix of different samples (large iLISI, Supplementary Fig. S1a, c, bottom panel).

We then evaluated the performance of estimating the embeddings induced by neighboring microenvironments in PRECAST. The estimated embeddings due to neighboring microenvironments were highly correlated with the underlying truth (Fig. 1c, middle panel; Supplementary Fig. S1d, top panel), suggesting PRECAST was able to well recover the spatial dependence of spots with microenvironments. Furthermore, the canonical correlation coefficients decreased as the scales in batch effects became large. Next, we evaluated the performance of the models in obtaining embeddings for biological effects between cell/domain types. For the average canonical correlation between the estimated aligned embeddings and the true latent features, PRECAST ranked at the top in the majority of scenarios (Supplementary Fig. S1b, top panel; Supplementary Fig. S1d, middle panel). This suggested that the aligned embeddings estimated by PRECAST were more accurate. We also showed that Pearson's correlation coefficients between the observed expression and the estimated cell/domain labels conditioned on the aligned embeddings from PRECAST were lower than those from other methods, except for scenario 3 (Supplementary Fig. S1b, d, bottom panel), suggesting PRECAST captured more relevant information regarding cell/domain clusters and, thus, facilitated downstream analysis.

Last, we compared the clustering performance of each method. As a unified method, PRECAST simultaneously estimates embeddings for biological effects between cell/domain types and cluster labels. For the other methods, we sequentially performed spatial clustering based on each of the estimated embeddings using SC-MEB, except for Seurat V3, which has its own clustering pipeline based on Louvain. PRECAST achieved the highest adjusted Rand index (ARI) and normalized mutual information (NMI) in all considered scenarios (Fig. 1c, bottom panel, and Supplementary Fig. S2a, b), while the other methods, such as Seurat V3 and Harmony, were sensitive to the data generation process. Moreover, we observed that all methods correctly chose the number of clusters. Notably, when using embeddings from PRECAST, other clustering methods such as SC-MEB, BASS, BayesSpace, and Louvain achieved comparable clustering performance to PRECAST (Supplementary Fig. S3a). PRECAST was also computationally efficient, exhibiting linear computational complexity with respect to the number of genes and the total number of spots (Supplementary Fig. S3b). It only took ~6 h to analyze a dataset with 2000 genes and 600,000 spots for a fixed number of clusters ($K = 7$; Supplementary Fig. S3b, left panel).

## Application to human dorsolateral prefrontal cortex Visium data

We applied PRECAST and the other methods to the analysis of four published datasets obtained via either Visium, ST, or Slide-seqV2 technologies (see "Methods"). By obtaining the estimated aligned embeddings and cluster labels from PRECAST, we could perform many downstream analyses using all tissue slides. Here, we showcase the differential expression (DE) analyses across detected domains, spatial variation analysis (SVA) adjusting for aligned embeddings as covariates, and trajectory inference/RNA velocity analysis. To examine the clustering performance with low-resolution data, we performed deconvolution analysis to infer the cell compositions of the domains detected by PRECAST.

To quantitatively show that PRECAST outperforms existing data integration methods, we first analyzed LIBD human DLPFC data generated using 10x Visium[21] that contained 12 tissue slices from three adult donors, comprising four tissue slices from each donor. In all 12 tissue slices, the median number of spots was 3844, and the median number of genes per spot was 1716. The original study provided manual annotations for the tissue layers based on the cytoarchitecture that allowed us to evaluate the performance of both the data integration and accuracy of spatial domain detection by taking the manual annotations as ground truth. For each method, we summarized the inferred embeddings for biological effects between cell/domain types using three components from either tSNE or UMAP and visualized the resulting tSNE/UMAP components with red/green/blue (RGB) colors in the RGB plot (Fig. 2a, right-top panel, and Supplementary Figs. S4a, b to S6a, b). The resulting RGB plots from PRECAST showed the laminar organization of the human cerebral cortex, and PRECAST provided smoother transitions across neighboring spots and spatial domains than those from other methods. For each of the other methods, we further performed clustering analysis to detect spatial domains using different methods with the inferred embeddings (Fig. 2a, right-bottom panel, and Supplementary Figs. S4c–S6c). We observed that the results from PRECAST had stronger laminar patterns and the estimated aligned embeddings carried more information about the domain labels (Supplementary Fig. S7a).

A unique feature of PRECAST is its ability to estimate slide-specific embeddings capturing spatial dependence in the neighboring cells/spots due to various neighboring microenvironments in different regions. Supplementary Fig. S8 provides RGB plots of the inferred embeddings for spatial dependence using three components from either tSNE or UMAP. We observed that spots in the domain of white matter had similar microenvironments, while spots in layers 1 to 6 had two distinct microenvironment patterns from left to right, suggesting potentially distinct functions in the left and right regions of layers 1 to 6.

PRECAST can offer outstanding data visualization compared with other methods. We visualized the inferred embeddings for biological effects between cell/domain types using two components from tSNE for each method (Fig. 2b and Supplementary Fig. S7b). The tSNE plots for PRECAST show that spots from different slices were well mixed

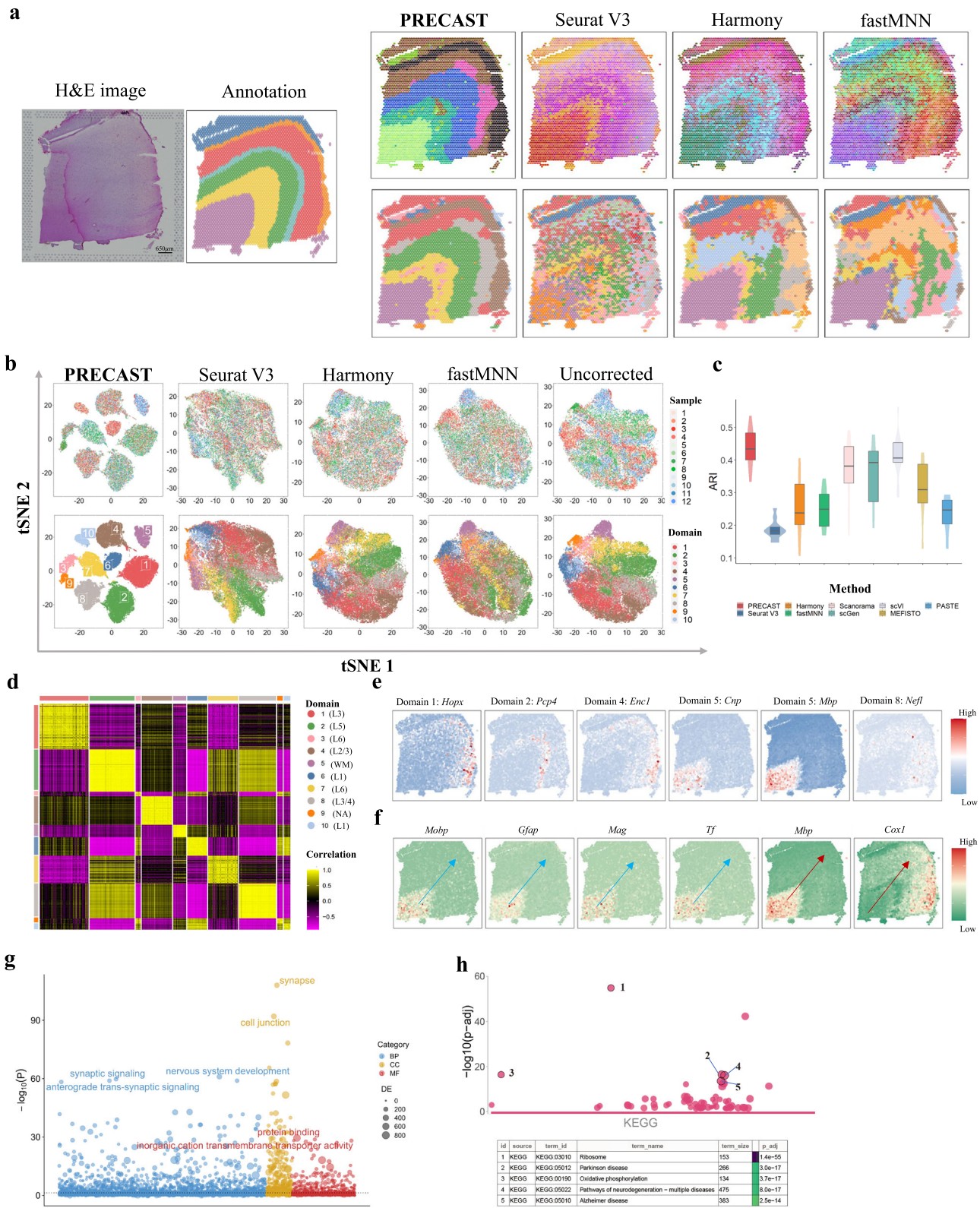

(Fig. 2b, top panel) while the domain clusters were well segregated (Fig. 2b, bottom panel), and there were significant improvements in visualization in comparison to the other methods, including the method applied with no corrections. Supplementary Fig. S7c shows that PRECAST achieved the best data integration in terms of F1 scores, iLISI, and cLISI. To evaluate the clustering accuracy, we used both ARI and NMI. As shown in Fig. 2c and Supplementary Fig. S7d, PRECAST achieved the highest ARI and NMI for the separate evaluation and combined evaluation: the median ARI was 0.434 for PRECAST, 0.382 for Scanorama, and 0.406 for scVI in the separate evaluation; and the ARI was 0.374 for PRECAST, 0.216 for Scanorama, and 0.301 for scVI in the combined evaluation. Using embeddings aligned from PRECAST, we further demonstrated that other clustering methods could achieve a similar clustering performance to PRECAST (Supplementary Fig. S9). A heatmap of Pearson's correlation coefficients among the detected domains shows the good separation of the estimated aligned

**Fig. 2 | Analysis of human DLPFC data ($n$ = 47,680 locations over 12 tissue sections). a** Left panel: H& E image and manual annotation of sample ID151674. Top panel: UMAP RGB plots of sample ID151674 for PRECAST, Seurat V3, Harmony, and fastMNN. Bottom panel: Clustering assignment heatmaps for these four methods. Color scheme used in clustering assignment heatmap for PRECAST is the same with (**b**) and (**d**). **b** tSNE plots for these four data integration methods with right-most column showing the analysis without correction; domains are labeled as in (**d**). **c** Box/violin plot of ARI values for PRECAST and other methods; SC-MEB was used in the other methods for clustering based on their aligned embeddings. In the box-plot, the center line and box lines denote the median, upper, and lower quartiles, respectively. **d** Heatmap of Pearson's correlation coefficients among detected embeddings across domains (Fig. 2d) and the correlations between domains. L1-L6, Layer 1–Layer 6; WM, white matter; NA, undetermined. **e** Spatial expression patterns of DE genes for Domain 1 (*HOPX*), Domain 2 (*PCP4*), Domain 4 (*ENC1*), Domain 5 (*CNP*), Domain 5 (*MBP*), and Domain 8 (*NEFL*) for sample ID151674. **f** Spatial expression patterns of genes associated with pseudotime: *MOBP*, *GFAP*, *MAG*, *TF*, *MBP*, and *COX1*, where the arrow represents the direction of the increased pseudotime. **g** Bubble plot of −log10(*p*-values) for GO enrichment analysis of genes associated with pseudotime. The *p*-values are based on one-sided hypergeometric tests without multiple testing adjustment. **h** Bubble plot of −log10(*p*-values) for KEGG enrichment analysis of SVGs while adjusting domain-relevant aligned embeddings by PRECAST for sample ID151674. The *p*-values are based on one-sided Fisher's exact tests with the Benjamini-Hochberg FDR corrections.

embeddings across domains (Fig. 2d) and the correlations between deeper layers were high, e.g., there were high correlations between layers 5 and 6, while correlations among the separated layers were low.

A key benefit of PRECAST is its ability to estimate aligned embeddings for biological effects between cell/domain types and joint labels for all slides. We performed DE analysis for the combined 12 slices (see "Methods"). In total, we detected 1331 DE genes with adjusted *p*-values of less than 0.001 among the 10 spatial domains identified by PRECAST, with 314 genes being specific to Domain 5, which corresponded to white matter (Supplementary Data 1). Many of these genes were reported to be enriched in different layers of DLPFC, i.e., *PCP4* (Domain 2, layer 5)[19], *HOPX* (Domain 1, layer 1/3), and *ENC1* (Domain 4, layer 2/3)[33] (Fig. 2e and Supplementary Figs. S10–S12). Next, we performed trajectory inference using the aligned embeddings and domain labels estimated by PRECAST (Supplementary Fig. S13). The pseudotime analysis inferred using the aligned embeddings from PRECAST was "sample-invariant" compared with that using embeddings from either PCA or DR-SC[34] for a single slide (Supplementary Fig. S14). In total, we identified 858 genes associated with the estimated pseudotime with adjusted *p*-values of less than 0.001. Among them, 373 were identified as DE genes in at least one domain (Fig. 2f and Supplementary Data 2). We further found that the pseudotime-associated genes identified by PRECAST were significantly enriched for nervous system development (Fig. 2g). Among the most enriched of these genes was *GFAP*, which encodes glial fibrillary acidic protein and plays an important role in human brain development[35].

To further show that the estimated embeddings for biological effects between cell/domain types from PRECAST were well aligned across tissue slides, we performed SVA analysis with the aligned embeddings from PRECAST as covariates for each slice to identify spatially variable genes (SVGs) with nonlaminar patterns. A detailed list of the genes identified at a false discovery rate (FDR) of 1% is available in Supplementary Data 3. Interestingly, many of the identified genes were related to immune function. For example, *ISG15* encodes a ubiquitin-like molecule induced by type I interferon, and ISG15 deficiency increases antiviral responses in humans[36,37]. Many studies in the literature have highlighted the importance of immune-brain interactions to the development of many disorders of the central nervous system[38,39]. Further enrichment analysis showed the genes from each slice to be highly enriched in many common pathways, suggesting the embeddings for biological effects between cell/domain types were effectively aligned by PRECAST (Fig. 2h and Supplementary Figs. S15–S19).

To demonstrate the robustness of PRECAST, we applied different methods to select top genes as input. As presented in Supplementary Fig. S20, when using the top 2000 highly variable genes (HVGs) as input, we observed similar patterns in results from different data integration methods. Supplementary Fig. S21 confirms the robustness of PRECAST by using top genes identified by different methods, such as SPARK[40], SPARK-X[41], SpatialDE[42], and nnSVG[43], as input.

## Application to mouse liver ST data

We further applied PRECAST and other methods to analyze eight sections of wild-type adult, female mouse livers from the caudate and right liver lobes of three female mice using ST technology[44]. In all eight sections, the median number of spots was 640, and the average number of genes was 15,302. The original study provided manual annotations based on marker genes that allows us to evaluate the performance of both the data integration and accuracy of spatial domain detection by taking the manual annotations as ground truth. For each method, we summarized the clustering performance of each section and combined sections using both ARI and NMI (Fig. 3a, top panel, and Supplementary Fig. S22a). PRECAST achieved the highest ARI and NMI in both cases: in each separate section, the median ARI was 0.24 for PRECAST, 0.18 for Seurat V3, and 0.02 for scVI; and jointly, in the combined sections, the value of ARI was 0.23 for PRECAST, 0.18 for Seurat V3, and 0.02 for scVI. We visualized the cluster labels obtained by PRECAST and other methods as well as the manual annotations (Supplementary Fig. S23) and found PRECAST performed best for each individual sample. On the other hand, PRECAST achieved better data integration than most of the other methods in terms of F1 score, iLISI, and cLISI (Fig. 3a, bottom panel, and Supplementary Fig. S22b) with comparable conditional correlations (Supplementary Fig. S22c). The tSNE plots for PRECAST show that spots from different sections were well mixed (Fig. 3b and Supplementary Fig. S22d, top panel), while the domain clusters were well segregated (Fig. 3b and Supplementary Fig. S22d, bottom panel), and there were significant improvements in visualization over other methods, including the method applied with no corrections. We further visualized spatial dependence due to variations in neighboring microenvironments using RGB plots of the inferred slide-specific embeddings (Supplementary Fig. S22e, f) and observed various patterns of microenvironments in the different sections. Using embeddings aligned from PRECAST, we further demonstrated that (spatial) clustering methods could achieve comparable clustering performance to PRECAST (Supplementary Fig. S24).

As a key feature, PRECAST estimates aligned embeddings for biological effects between cell/domain types and joint labels for the combined sections. We performed DE analysis of the combined sections (see Methods). In total, we detected 367 DE genes with adjusted *p*-values of less than 0.001 in all seven spatial domains detected by PRECAST (Supplementary Data 4). A heatmap of the findings shows the good separation of the DE genes across different spatial domains (Fig. 3c). Many of these genes are markers that define particular cellular regions in liver lobes, i.e., *Cyp2e1*, *Cyp2c37*, *Oat*, and *Slc1a2* for central veins (Domains 1–2)[44,45]; *Cyp2f2*, *Hal*, *Sds*, and *Ctsc* for portal veins (Domains 6-7)[44]; and *Gsn*, *Vim*, and *Col3a1* for the mesenchyme (Domain 5)[44]. Further enrichment analysis shows that genes specific to both central veins and portal veins were highly enriched for metabolic processes, with central veins that were more enriched for fatty acid metabolism, while portal veins that were more enriched for amino acid metabolism (Supplementary Fig. S25). By performing enrichment analysis for DE genes unique to each of two subtypes in central/portal

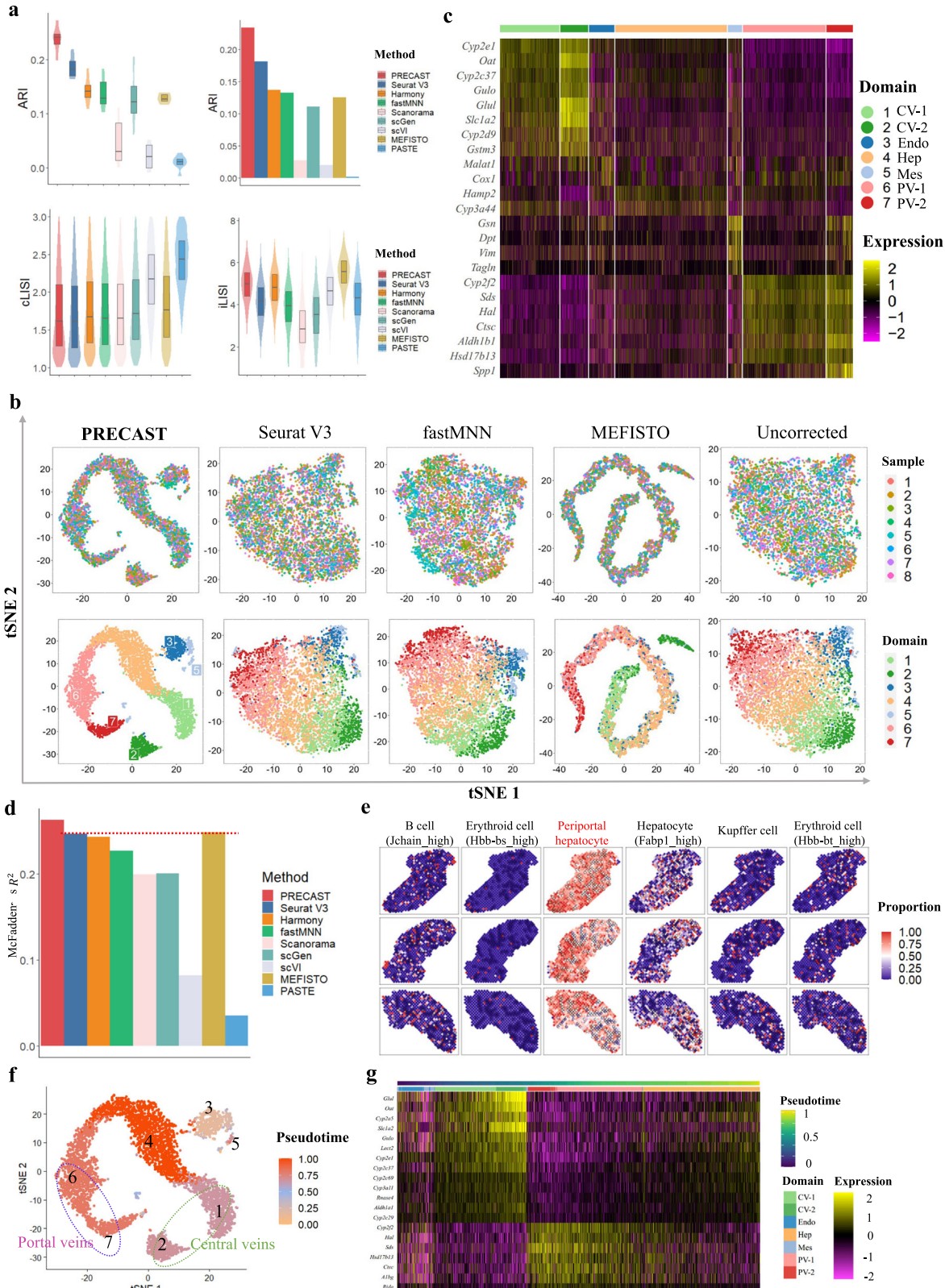

veins, we found pathways unique to each subtype of central/portal veins (Supplementary Fig. S26).

To examine the cell compositions of each spatial domain detected by PRECAST, we performed cell-type deconvolution analysis of all mouse liver datasets using scRNA-seq data in the Mouse Cell Atlas (MCA)[46] as the reference panel. To assess the performance in spatial clustering, we evaluated the associations between the cell type proportions obtained by cell-type deconvolution and the cluster labels estimated by PRECAST and other methods. The results displayed in Fig. 3d suggest that PRECAST retained the largest MacFadden's adjusted $R^2$ than the other methods. In addition, cell-type deconvolution enabled us to spatially map 17 cell types annotated in the MCA dataset for liver tissue sections (Fig. 3e and Supplementary Figs. S27–S28). We observed a high proportion of estimated values for

**Fig. 3 | Analysis of mouse liver ST data (n = 4865 locations over 8 tissue sections). a** Top panel: Box/violin plot of ARI values of each sample for PRECAST and the other methods (left); bar plot of ARI value of combined samples for PRECAST and other methods (right). Bottom panel: Box/violin plots of cLISI and iLISI values for PRECAST and other methods. Color scheme of each method is the same as in (**d**). In the boxplot, the center line and box lines denote the median, upper, and lower quartiles, respectively. **b** tSNE plots for four data integration methods, with the right-most column showing analysis without correction. Color scheme of each domain is the same as in (**c**) and (**g**). **c** Heatmap of differentially expressed genes for each domain identified by PRECAST. CV-1, central veins 1; CV-2, central veins 2; Endo, endothelial cells; Hep, hepcidin-related cells; Mes, mesenchymal-related

cells; PV-1, portal veins 1; PV-2, portal veins 2. **d** Bar plot of McFadden's adjusted $R^2$ values for PRECAST and other methods. McFadden's adjusted $R^2$ measures the association between the cluster label obtained by each method and the cell proportion obtained by RCTD cell-type deconvolution, and a larger value indicates a stronger association. **e** Visualization of the cell type proportions mapped to spatial coordinates for six cell types of the first three samples. **f** Visualization of the combined trajectory inferred by PRECAST in tSNE plot of all samples. Domains 1-2 and Domains 6-7 representing central veins and portal veins, respectively, are circled. **g** Heatmap for genes with expression change in the Slingshot pseudotime inferred by PRECAST.

periportal and pericentral hepatocytes, which is consistent with the findings in the existing literature[44].

Lastly, we performed trajectory inference using aligned embeddings and estimated domain labels from PRECAST, to examine the cell lineages in the detected domains. Figure 3f shows the inferred pseudotime mapped to PRECAST-induced tSNE, which suggests the central veins differentiated earlier than portal veins, in accordance with findings that showed Wnts and R-spondin3 signals were released from the central veins and transited along the venular wall towards the perivenous hepatocytes[45]. Based on the inferred pseudotime, we identified differentially expressed genes along the cell pseudotime using TSCAN[47]. The heatmap of the expression of the top 20 most significant genes (Fig. 3g) suggested the occurrence of some interesting dynamic expression patterns over pseudotime.

We further confirmed the robustness of PRECAST by using top genes identified by different methods as input (Supplementary Figs. S29 and S30).

## Application to mouse olfactory bulb Slide-seqV2 data

In Visium Spatial Gene Expression, barcoded beads (55 μm diameter) with a center-to-center distance of 100 μm are used to capture mRNA[11]. The Slide-seq technique was developed to perform for high-resolution SRT using 10-μm-diameter barcoded beads[9,10], and Slide-seqV2 further improved the detection sensitivity. To show the scalability of PRECAST, we analyzed a mouse OB dataset generated using Slide-seqV2 technology[22]. In this dataset, spatial transcriptomic information was obtained from a total of 20 OB sections distributed evenly along the anterior-posterior axis. We removed four slides due to the quality of the sections from the end of the mouse OB region, and analyzed 16 containing 21,571 genes, on average, from over a total of 693,863 spots using PRECAST and other methods. After quality control (QC), we obtained data of a lower resolution (~4000 spots) by collapsing nearby spots in each slide (see "Methods"), and the resulting resolution was similar to that of Visium (average of 10 cells per spot). To further examine the structure of the mouse OB, we relied on the structural annotations in the Allen Brain Atlas[48] (Fig. 4a). With near-single-cell resolution, the identified aligned embeddings and cluster labels both showed more fine-scale cell-type distribution patterns in the mouse OB[49]. In comparison, when we lowered the resolution, PRECAST estimated the aligned embeddings and cluster labels with smoother spatial patterns at the expense of less detailed local spatial information (Fig. 4b and Supplementary Figs. S31–S34). Moreover, the resulting RGB plots from PRECAST showed the laminar organization of the mouse OB, and PRECAST provided smoother transitions across neighboring spots and spatial domains than the other methods (Supplementary Figs. S31a, b to S34a, b). We next visualized the inferred aligned embeddings for biological effects between cell/domain types using two components from tSNE from each method (Fig. 4c and Supplementary Fig. S35) and showed that, using PRECAST, the spatial spots were mixed well across the 16 slides, while the cell/domain clusters were well segregated, suggesting PRECAST was more effective at spatial data integration. We further visualized the inferred slide-specific embeddings for spatial dependence due to variations in the

microenvironment using RGB plots (Supplementary Fig. S36) and observed a few microenvironmental patterns in the inner layers (e.g., the granule cell layer, GCL), middle layers (e.g., the glomerular layer, GL), and outer layers (e.g., the olfactory nerve layer, ONL).

At reduced resolution, PRECAST detected 12 spatial domains with laminar organization, including the rostral migratory stream (RMS, Domain 1), GCL (Domain 2), GCL/inner plexiform layer (GCL/IPL, Domain 3), mitral cell layer (MCL, Domain 4), outer plexiform layer (OPL, Domain 5), GL (Domains 6 and 7), and ONL (Domains 8), with Domains 9–12 belonging to low-quality regions or experiment artifacts. To characterize the transcriptomic properties of the spatial domains identified by PRECAST, we performed DE analysis of the combined 16 tissue slides (see "Methods"). In total, we detected 4131 DE genes with adjusted $p$-values of less than 0.001 in all 12 spatial domains detected by PRECAST (Supplementary Data 5), including representative genes that define particular cellular layers in the mouse OB, e.g., *Sox2ot* and *Sox11* (RMS)[50,51]. The heatmap of the findings shows the good separation of the DE genes across different spatial domains (Supplementary Fig. S37), and we found that genes specific to Domain 1 (RMS) were enriched for myelin sheath and structural constituents of myelin sheath (Supplementary Fig. S38). Compared with the reduced-resolution data, when near-single-cell level resolution was used, PRECAST identified 24 cell clusters, including fine-scale cell-type clusters. To better visualize each detected cell cluster, we plotted a heatmap of each cluster assignment for all 16 slides (Supplementary Figs. S39–S40). A heatmap of DE genes across different cell clusters showed Clusters 1–3 were subtypes of granule cells, and Clusters 4-6 belonged to cells in MCL and GL (Supplementary Fig. S41).

To examine the cell compositions in each spatial domain detected by PRECAST at the reduced resolution, we performed cell-type deconvolution analysis of all 16 tissue slides using scRNA-seq data from adult mouse OB as the reference panel[52]. As shown in Fig. 4d and Supplementary Fig. S42a, Domain 1 (RMS) was enriched for immature neurons. Immature neurons reportedly migrate to the OB through RMS[53]. Unsurprisingly, we found that Domains 2–3 (GCL) were dominated by two primary subtypes of granule cell, with a larger proportion of immature neurons in Domain 2 (inner) and an enrichment of mitral and tufted cells in Domain 4 (MCL). Domain 8 (ONL) was primarily enriched in olfactory sensory neurons (OSNs); OSNs express odorant receptors in the olfactory epithelium[54]. We additionally quantified the association between the inferred cell type proportions and the domain labels estimated by PRECAST and the other methods using MacFadden's adjusted $R^2$ (Fig. 4e, top panel), and PRECAST achieved the highest $R^2$. Then, we manually annotated each spot with the cell type present at the highest proportion[55] and quantitatively evaluated the clustering performance of PRECAST and other methods. As shown in Fig. 4e (bottom panel) and Supplementary Fig. S42b, c, PRECAST achieved the highest ARI and NMI values assessed separately for each slide or jointly for the combined slides. Further analyses showed that PRECAST achieved better data integration, with the highest iLISI and the lowest cLISI (Supplementary Fig. S42d), and maintained comparable conditional Pearson's correlations with the other methods (Supplementary Fig. S42e).

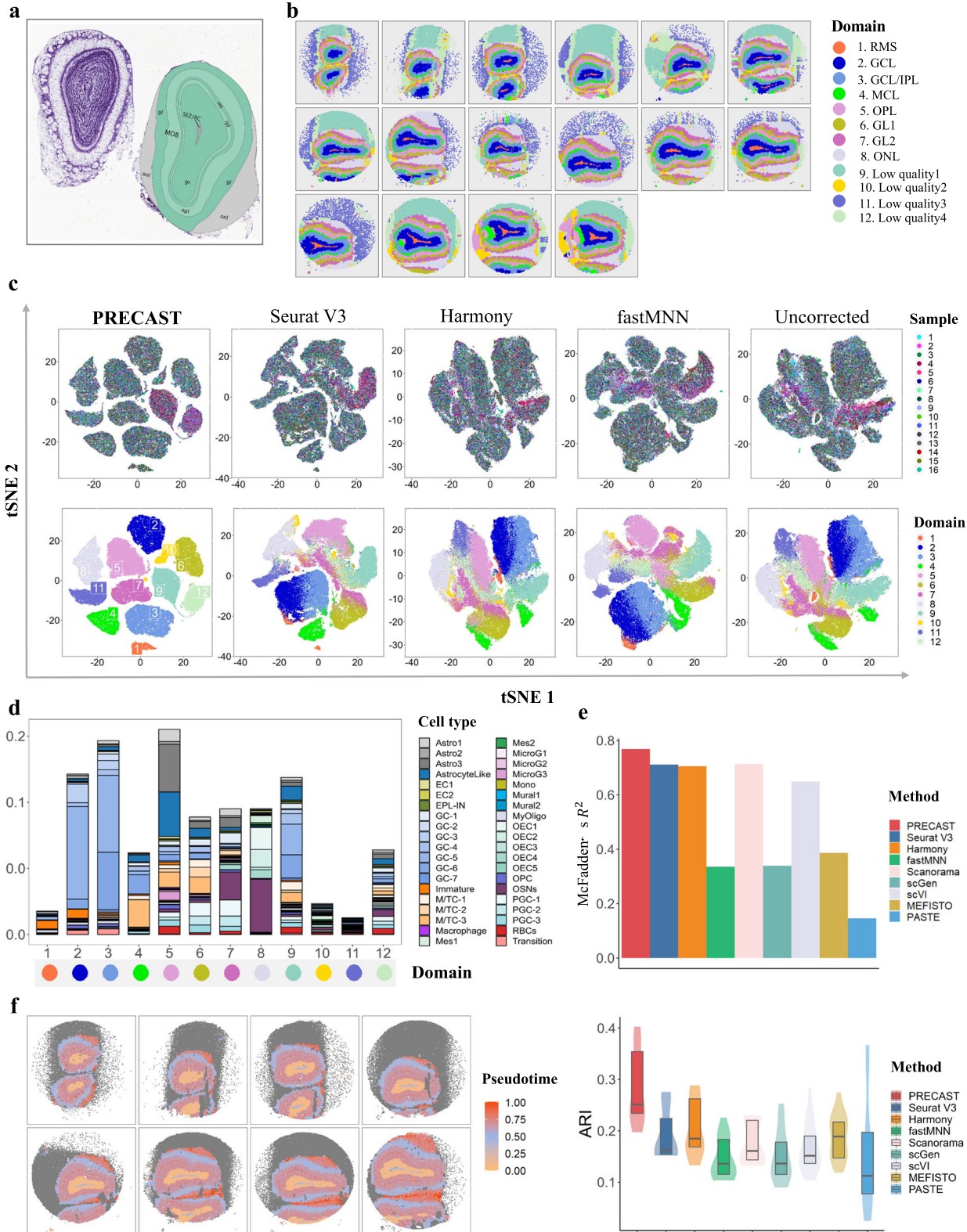

We performed trajectory inference to examine the cell lineages among the detected domains using aligned embeddings and estimated domain labels from PRECAST. In general, the estimated trajectory showed an "inside-out" sequence, consistent with the general under-standing that OB neurons migrate from the subventricular zone along the RMS to the OB, before migrating radially out of the RMS in an inside-out sequence[56] (Fig. 4f and Supplementary Fig. S43a). Further DE analysis identified genes along the pseudotime (Supplementary Fig. S44b).

## Application to hepatocellular carcinoma Visium data

To study the dynamics of tumorigenesis in tumors and tumor-adjacent tissues, we further analyzed four slides of in-house HCC data generated using the 10x Visium platform, with two slides from tumors (HCC1 and HCC2) and two from tumor-adjacent tissues (HCC3 and HCC4) from an HCC patient. The median number of spots was 2748, and the median number of genes per spot was 3635. Figure 5a shows a histology image (top panel) with manual annotations for tumor/normal epithelium

**Fig. 4 | Analysis of mouse olfactory bulb data (*n* = 594,890 locations over 16 tissue sections). a** Structure of the mouse olfactory bulb annotated using the Allen Brain Atlas. **b** Clustering assignment heatmaps for 16 tissue slides by PRECAST, where the first row shows samples 1–6, the second row samples 7–12, and the last row samples 13–16 (RMS, rostral migratory stream; GCL, granule cell layer; IPL, inner plexiform layer; MCL, mitral cell layer; OPL, outer plexiform layer; GL, glomerular layer; ONL, olfactory nerve layer). Color scheme for domains detected in PRECAST is as in (**c**) and (**d**), and the order of domain labels in (**b**) is the same as in (**c**), (**d**), and (**e**). **c** tSNE plots for four data integration methods with the right-most column showing analysis without correction. **d** Percentage of different cell types in each domain detected by PRECAST with scaling. **e** MacFadden's adjusted $R^2$ between the inferred cell type proportions and the estimated domain labels by PRECAST and other methods (top panel); boxplot of ARI values of 16 samples for PRECAST and other methods, where each spot is annotated using the cell type, with the highest proportion from the spatial deconvolution (bottom panel). In the boxplot, the center and box lines denote the median, upper, and lower quartiles, respectively. (**f**) Visualization of the trajectory inferred by PRECAST in spatial heatmap for samples 1–8.

(TNE) and stroma provided by a pathologist (bottom panel). Consistent with the DLPFC data, the RGB plots generated by PRECAST clearly segregated the tissue slices into multiple spatial domains (Fig. 5b, top panel), with neighboring spots and spots in the same domain across multiple slides more closely sharing similar RGB colors than those generated by other methods (Supplementary Fig. S44a, b). Similarly, the PRECAST spatial heatmaps of clustering assignment for the four tissue slides resembled the corresponding RGB plots and presented more spatial patterns across the tissue slides than those from the other methods (Fig. 5b, bottom panel; Supplementary Fig. S44c). We further visualized the inferred embeddings for biological effects between cell/domain types using two components from tSNE for each method (Fig. 5c and Supplementary Fig. S45a), in which the tSNE plots for PRECAST showed that spots from different slices were mixed well (Fig. 5c, top panel) while the domain clusters were well segregated (Fig. 5c, bottom panel), with significant improvements in visualization. A heatmap of Pearson's correlation coefficients among the detected domains shows the good separation of estimated embeddings across domains (Supplementary Fig. S45b), in which correlations between regions in TNE were high, and correlations between regions for TNE and stroma were low. We further visualized the spatial dependence due to microenvironment variations using RGB plots of the inferred slide-specific embeddings (Supplementary Fig. S46a) and observed variations in microenvironmental patterns between Domains 1 and 3, and between Domains 4 and 5.

To characterize the transcriptomic properties of the spatial domains identified by PRECAST, we performed DE analysis of the combined four tissue slides (see Methods). In total, we detected 2093 DE genes with adjusted *p*-values of less than 0.001 in all nine spatial domains detected by PRECAST, with 539 genes being specific to Domains 1–5, which corresponded to TNE regions (Supplementary Data 6). A heatmap and ridge plots of the findings showed the good separation of the DE genes across different spatial domains (Supplementary Figs. S46b, S47–S48). In TNE regions (Domains 1–5), we further found that genes specific to Domains 1 and 5 were highly enriched in pathways of chemical carcinogenesis DNA adducts and chemical carcinogenesis receptor activation. Genes specific to Domain 4 were enriched in signaling pathways of RAF1 mutants and signaling by RAS mutants, and genes specific to Domains 2 and 3 were highly enriched in complement and coagulation cascade pathways (Supplementary Fig. S49). Genes specific to Domains 6–9 were enriched in the angiogenesis pathway, and we found 40 out of 427, 34 out of 606, 25 out of 281, and 18 out of 240 angiogenesis signature genes, respectively, in Domains 6–9, from multiple studies[57–60]. Interestingly, Domains 1–3 were only present in the tumor tissues (HCC1 and HCC2), Domain 5 was only present in the tumor-adjacent tissues (HCC3 and HCC4), while Domain 4 was shared across the tumor and tumor-adjacent tissues.

To identify SVGs other than those that were merely relevant to domain differences, we performed SVA analysis with the embeddings estimated by PRECAST as covariates for each slice. A detailed list of genes identified at an FDR of 1% is available in Supplementary Data 7. By performing functional enrichment analysis of these SVGs, we detected SVGs adjusted for domain-relevant covariates to be highly enriched in many common pathways in the four HCC slices, e.g.,

cytoplasmic translation, and cytosolic ribosome (Supplementary Figs. S50–S51).

Next, to examine the cell compositions of each spatial domain detected by PRECAST, we performed cell-type deconvolution analysis of all four HCC slides using scRNA-seq data as the reference panel (see Methods). The scRNA-seq reference panel consisted of malignant and tumor microenvironment cells, including cancer-associated fibroblasts (CAFs), tumor-associated macrophages (TAMs), tumor-associated endothelial cells (TECs), cells of an unknown entity but expressing hepatic progenitor cell markers (HPC-like), and immune cells[61]. As shown in Fig. 5d, e and Supplementary Fig. S52a, the proportions of malignant cells were substantially higher in Domains 1–5, while HPC-like cells were seen at higher proportions in Domain 7. In Domain 6, we observed an increased proportion of TAMs and immune cells and genes specific to this domain included *TGFB1* and *MMP2*, which have been used for the classification of TAMs[62,63].

The estimated aligned embeddings and cluster labels can also be used in RNA velocity analysis to investigate the directed transcriptional dynamics of tumorigenesis when spliced and unspliced mRNA are available (see "Methods"). Interestingly, two cell lineages were identified, with one that originated in Domain 2 (TNE) and spread to Domains 4 and 5 (TNE), followed by Domains 6, 7, and 9 (stroma), and the other that originated in Domains 1 and 3 (TNE) and spread to Domain 8 (stroma). The TNE regions in Domain 2 residing in HCC1 may play a key role in tumorigenesis with Domain 4 (TNE) shared among the four slides and Domain 5 (TNE) in the tumor-adjacent tissues (Fig. 5f and Supplementary Fig. S52b). To infer cell states in the identified TNE, we further performed RNA velocity analysis using the spots identified in Domains 1–5. An expression heatmap of the top genes associated with cell states with induction close to 0 and repression close to 1 is shown in Fig. 5g (see "Methods"). The TNEs in Domains 1 and 2 tended to be transcriptionally active, while TNEs in Domains 4 and 5 tended to be repressed and show no transcription. The top genes associated with cell states included *SPINK1*, *RPL30*, and *IL32*, which highlights the importance of genes associated with cell states in HCC[64–66].

## Discussion

PRECAST takes, as input, matrices of normalized expression levels and the physical location of each spot across multiple tissue slides. The output of PRECAST comprises all aligned embeddings for cellular biological effect, slide-specific embeddings that capture spatial dependence in neighboring cells/spots, and estimated cluster labels. In contrast to other existing methods of data integration, PRECAST is a unified and principled probabilistic model that simultaneously estimates embeddings for cellular biological effects, performs spatial clustering, and more importantly, aligns the estimated embeddings across multiple tissue sections. Thus, we recommend applying PRECAST first before a comprehensive data analysis pipeline is deployed. By applying PRECAST, the aligned embeddings and estimated cluster labels can be used for many types of downstream analyses, such as visualization, trajectory analysis, and SVA and DE analysis for combined tissue slices. In more detail, we developed a module to further remove batch effects across multiple tissue slides based on

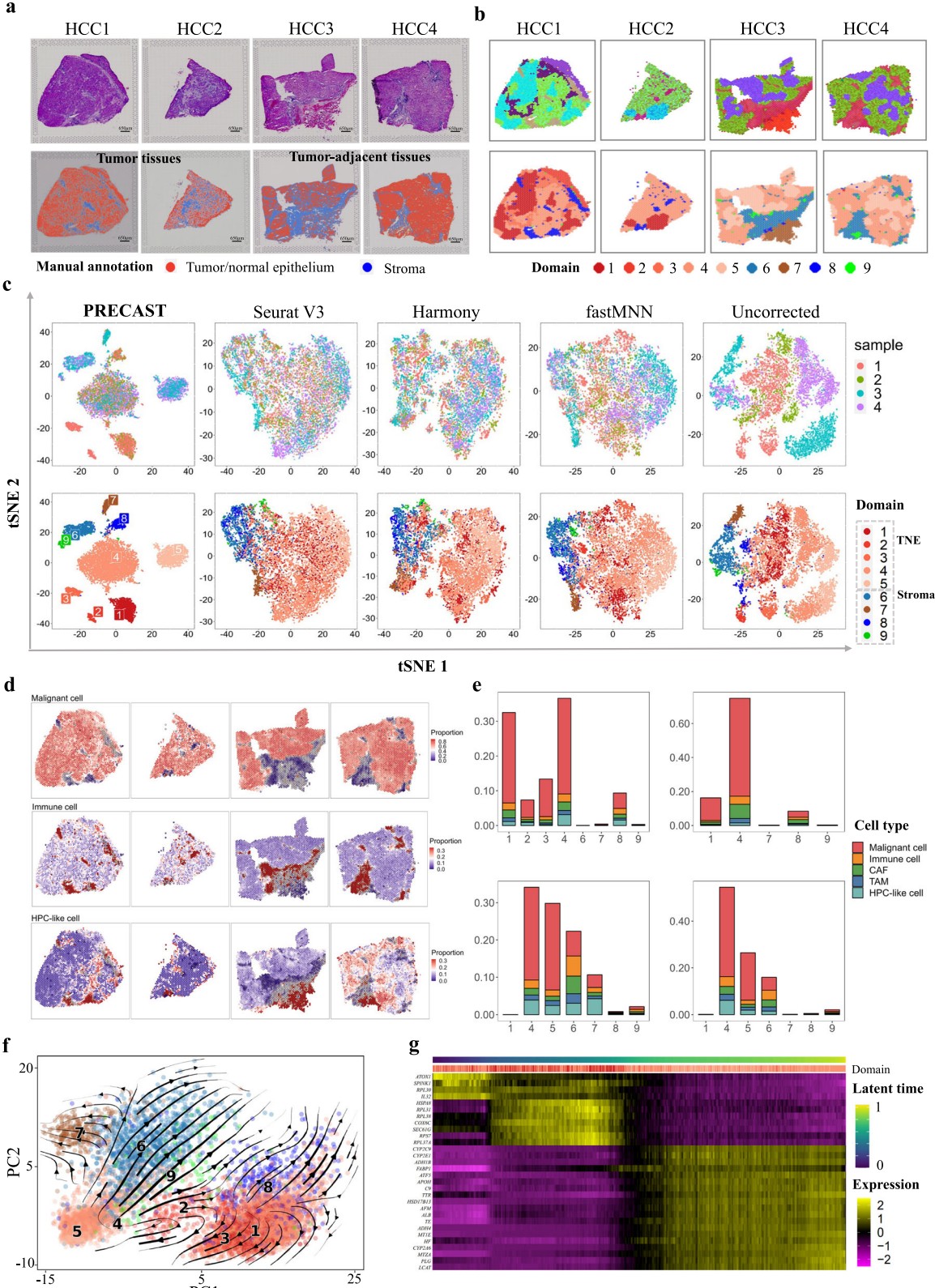

**Fig. 5 | Analysis of data for four human HCC sections. a** Top panel: H& E images from four tissue slides. Bottom panel: Manual annotation by a pathologist of four tissue slides. **b** Top panel: UMAP RGB plots of PRECAST for four tissue slides. Bottom panel: Clustering assignment heatmaps for four tissue sections by PRE-CAST. Color scheme for clustering assignment heatmap in PRECAST is the same as in (**c**), (**f**), and (**g**). **c** tSNE plots for four data integration methods with the right-most column showing analysis without correction; domains are labeled as in (**e**) and (**f**). TNE, tumor/normal epithelium. **d** Spatial heatmap of deconvoluted cell proportions in malignant cells, immune cells, and HPC-like cells. **e** Percentage of different cell types in each domain detected by PRECAST, with scaling to the summation of all cell types across all domains equal to 100%. **f** PC plot of estimated RNA velocity. **g** Heatmap of genes with expression change in latent time.

housekeeping genes, making expression data comparable for different cell/domain clusters. This module is also applicable to the examination of expressional differences caused by multiple conditions when such information on tissue slides is obtainable.

PRECAST simultaneously performs dimension reduction and spatial clustering while using simple projections to align embeddings, and uniquely estimating embeddings that capture spatial dependence of neighboring cells/spots due to varied microenvironments. Recently, Liu et al.[34] showed that, compared with methods that perform dimension reduction and spatial clustering sequentially[18–20], joint methods can estimate embeddings for cellular biological effects more efficiently while accounting for the uncertainty in obtaining low-dimensional features from sequential analysis. A similar strategy has been described in previous self-supervised learning literature[67,68]. PRECAST also takes advantage of CAR to account for the local microenvironments of neighboring spots, and an intrinsic CAR component has been used to promote spatial smoothness in the observed expressions of SRT data[69]. We showed that, by projecting non-cellular biological effects onto cellular biological space, different sample slides exhibit a constant shift in the centroid of each cell/domain type. With the assistance of joint modeling, we can use a subclass of CAR, intrinsic CAR, to simultaneously account for both smoothness in neighboring embeddings and shift across batches due to non-cellular biological effects such as complex batch effects.

With the advent of high-throughput technologies for SRT, data integration is particularly relevant for analyzing SRT datasets from multiple tissue slides. Analysis of a single section with current state-of-the-art techniques, e.g., 10x Visium and Slide-seqV2, only covers a tiny area of the region of interest, and it takes a few or dozens of slides to cover the whole tissue/organ. In this case, biological variations between cell/domain types are often confounded by factors related to data generation processes. Methods in data integration, which serve as the first step before downstream analyses, not only align the embeddings but also estimate the shared cell/domain clusters across samples[23–26]. Most existing methods of data integration were designed to analyze scRNA-seq data without considering additional spatial information in the SRT data.

We examined the SRT data generated by three major platforms, 10x Visium, ST, and Slide-seqV2, with different spatial resolutions. With 10x Visium, mRNA is captured in 55-$\mu$m-diameter spots, while Slide-seqV2 achieves a higher resolution with 10$\mu$m-diameter spots. As common sense implies, with low-resolution datasets, PRECAST recovers aligned embeddings for cellular biological effects and cluster labels with smoother spatial patterns while losing the detailed local spatial information. Whereas, at near-single-cell resolution, the identified aligned embeddings and cluster labels show more fine-scale cell-type distribution patterns. Using four datasets, we demonstrated that PRECAST can successfully perform data integration with aligned embeddings across tissue sections, such that spots across different tissue sections are well mixed while cell/domain clusters are well segregated, improve clutering performance, and detect varied microenvironments in tissues. When applied to an HCC dataset, PRECAST identified five spatial domains belonging to TNE cells that were consistent with both manual annotations and spatial deconvolution. We further performed RNA velocity analysis to show the potential velocity of spots in different regions, shedding light on the tumorigenesis in the context of the tissues. To demonstrate the scalability of PRECAST, we analyzed a Slide-seqV2 dataset of 16 slides equally distributed along the anterior-posterior axis of a mouse OB.

PRECAST provides opportunities for new exciting research routes. Firstly, when SRT datasets of single-cell resolution are available, its use can be extended to the integration of multimodal single-cell data. For example, integrating single-cell ATAC-seq will allow users to examine cell-type-specific regulatory mechanisms in the spatial context of tissues. Secondly, it would be interesting to integrate single-

cell-resolution SRT datasets with CITE-seq data, thus integrating spatial transcriptomics with immunophenotyping, in which surface proteins are detected by antibody-derived tags. This would enable the exploration of surface proteins not measured in SRT datasets.

## Methods
### PRECAST model
Here, we present a basic overview of PRECAST, and further details are available in the Supplementary Notes. PRECAST is a data integration method for SRT data from multiple tissue slides. The proposed method involves simultaneous dimension reduction and spatial clustering built on a hierarchical model with two layers, as shown in Fig. 1a. The first layer, the dimension-reduction step, relates gene expression to the shared latent embeddings, while the second layer, the spatial-clustering step, relates the shared latent embeddings and spatial coordinates to the cluster labels. In the dimension-reduction step, an intrinsic CAR model captures the spatial dependence induced by neighboring microenvironments in the low-dimensional embedding space, while in the spatial-clustering step, a Potts model promotes spatial smoothness in the cluster label space. Using simple projections of the batch effects and/or biological effects of slides onto the space of biological effects between cell/domain types, PRECAST aligns cell/domain clusters across multiple tissue slides with the shared distributions of embeddings of each cell/domain and detects their cell/domain labels. With $M$ SRT datasets, we observe an $n_r \times p$ normalized expression matrix $\mathbf{X}_r = (\mathbf{x}_{r1}, \cdots, \mathbf{x}_{ri}, \cdots, \mathbf{x}_{rn_r})^\top$ for each sample $r(=1,\cdots,M)$, where $\mathbf{x}_{ri} = (x_{ri1}, \cdots, x_{rip})^\top$ is a $p$-dimensional normalized expression vector for each spot $s_{ri} \in \mathbb{R}^2$ of sample $r$ on square or hexagonal lattices, among others; while the cluster label of spot $s_{ri}$, $y_{ri} \in \{1,\cdots,K\}$, and $q$-dimensional shared embeddings, $\mathbf{z}_{ri}$'s, are unavailable. Without loss of generality, we assume that, for each sample $r$, $\mathbf{x}_{ri}$ is centered, and PRECAST models the centered normalized expression vector $\mathbf{x}_{ri}$ with its latent low-dimensional feature, $\mathbf{z}_{ri}$, and class label, $y_{ri}$, as

$$\mathbf{x}_{ri} = \mathbf{W}(\mathbf{z}_{ri} + \mathbf{v}_{ri}) + \boldsymbol{\varepsilon}_{ri}, \boldsymbol{\varepsilon}_{ri} \sim N(\mathbf{0}, \Lambda_r), \tag{1}$$

$$\mathbf{z}_{ri} | y_{ri} = k \sim N(\mu_k, \Sigma_k), \tag{2}$$

where $\Lambda_r = \text{diag}(\lambda_{r1}, \cdots, \lambda_{rp})$ is a diagonal matrix for residual variance, $\mathbf{W} \in \mathbb{R}^{p \times q}$ is a loading matrix that transforms the $p$-dimensional expression vector into $q$-dimensional embeddings shared across $M$ datasets, $\mu_k \in \mathbb{R}^{q \times 1}$ and $\Sigma_k \in \mathbb{R}^{q \times q}$ are the mean vector and covariance matrix for the $k$th cluster, respectively, and $\mathbf{v}_{ri}$ is a $q$-dimensional slide-specific latent vector that captures the spatial dependence among neighboring spots and aligns embeddings across datasets. Equation (1) is related to the high-dimensional expression vector ($\mathbf{x}_{ri}$) in $p$ genes with a low-dimensional feature ($\mathbf{z}_{ri}$) via a probabilistic PCA model[70] with consideration of spatial dependence while Eq. (2) is a Gaussian mixture model (GMM)[71] for this latent feature among all spots across $M$ datasets. To promote spatial smoothness in the space of cluster labels, we assume each latent class label, $y_{ri}$, is interconnected with the class labels of its neighborhoods via a discrete hidden Markov random field (HMRF). In detail, we use the following Potts model[72] for the latent labels,

$$P(\mathbf{y}_r) = C_r(\beta_r)^{-1} \exp\left\{ -\frac{1}{2} \sum_i \sum_{i' \in N_{ri}} \beta_r (1 - \delta(y_{ri}, y_{ri'})) \right\}, \tag{3}$$

where $C_r(\beta_r)$ is a normalization constant that does not have a closed form, $N_{ri}$ is the neighborhood of spot $s_{ri}$ in sample $r$, and $\beta_r$ is the sample-specific smoothing parameter that captures the label similarity among the neighboring spots. However, we assume a continuous

multivariate HMRF for the vector, $\mathbf{v}_{ri}$, which captures spatial dependence in the embedding space. In detail, we assume an intrinsic CAR model[73] for $\mathbf{v}_{ri}$

$$\mathbf{v}_{ri}|\mathbf{v}_{[n_r]\setminus i} \sim N(\mu_{v_{ri}}, m_{ri}^{-1}\Psi_r), \quad (4)$$

where subscript $_{[n_r]\setminus i}$ denotes all spots but $s_{ri}$ in sample $r$, $m_{ri}$ is the number of neighbors of spot $i$ in sample $r$, $\mu_{v_{ri}} = m_{ri}^{-1}\sum_{i' \in N_{ri}} \mathbf{v}_{ri'}$ is the conditional mean relevant to the neighbors of that spot $s_{ri}$, and $\Psi_r$ is a $q \times q$ conditional covariance matrix for the elements of $\mathbf{v}_{ri}$. Conventionally, the joint distribution of the intrinsic CAR model is non-identifiable, as the mean of the joint distribution of intrinsic CAR is not zero. As shown in the next section, non-cellular biological effects, e.g., batch effects, from each slide can be projected onto the cellular biological space, which can be corrected by the non-zero-mean property of intrinsic CAR.

## Projections of non-cellular biological effects

For each tissue slide, we assume the normalized expressions in each sample $r$ can be decomposed into additional parts with respect to non-cellular biological effects as follows:

$$\mathbf{x}_{ri} = \mathbf{W}(\mathbf{z}_{ri} + \mathbf{v}_{ri}) + \mathbf{W}_r\boldsymbol{\zeta}_{ri} + \boldsymbol{\varepsilon}_{ri}, \quad (5)$$

where $\mathbf{v}_{ri}$ is a $q$-dimensional vector that captures the spatial dependence among neighboring spots; $\mathbf{W}_r \in \mathbb{R}^{p \times \tilde{q}}$ is a loading matrix for a factor related to non-cellular biological effects; and $\boldsymbol{\zeta}_{ri}$, independent of $(\mathbf{z}_{ri}, \mathbf{v}_{ri})$, is the corresponding $\tilde{q}$-dimensional vector. Assuming cell biological space ($\mathbf{W}$) and non-cellular biological space ($\mathbf{W}_r$) are non-orthogonal, we can project $\mathbf{W}_r$s onto the column spaces of $\mathbf{W}$, i.e., $\widehat{\mathbf{W}}_r = \mathbf{W}(\mathbf{W}^T\mathbf{W})^{-1}\mathbf{W}^T\mathbf{W}_r$, and then rewrite the normalized expressions as the following

$$\mathbf{x}_{ri} \approx \mathbf{W}(\mathbf{z}_{ri} + \mathbf{v}_{ri} + \widetilde{\mathbf{W}}^T\mathbf{W}_r\boldsymbol{\zeta}_{ri}) + \boldsymbol{\varepsilon}_{ri},$$

where $\widetilde{\mathbf{W}} = \mathbf{W}(\mathbf{W}^T\mathbf{W})^{-1}$. We denote $\mathbf{v}_{ri} = \mathbf{v}_{ri} + \widetilde{\mathbf{W}}^T\mathbf{W}_r\boldsymbol{\zeta}_{ri}$, $\mu'_{v_{ri}} = E(\mathbf{v}_{ri}|\mathbf{v}_{[n_r]/i})$, $m_{ri}^{-1}\Psi'_r = \text{var}(\mathbf{v}_{ri}|\mathbf{v}_{[n_r]/i})$, then the conditional mean and variance in the intrinsic CAR component can be written as

$$E(\mathbf{v}_{ri}|\mathbf{v}_{[n_r]/i}) = \mu'_{v_{ri}} + \widetilde{\mathbf{W}}^T\mathbf{W}_r E(\boldsymbol{\zeta}_{ri}) \equiv \mu_{v_{ri}},$$

$$\text{var}(\mathbf{v}_{ri}|\mathbf{v}_{[n_r]/i}) = m_{ri}^{-1}\Psi'_r + \widetilde{\mathbf{W}}^T\mathbf{W}_r\text{var}(\boldsymbol{\zeta}_{ri})\mathbf{W}_r^T\widetilde{\mathbf{W}} \equiv m_{ri}^{-1}\Psi_r,$$

where we assume $\text{var}(\boldsymbol{\zeta}_{ri}) = m_{ri}^{-1}\Psi''_r$. Then, the projected approximated model can be written as

$$\mathbf{x}_{ri} \approx \mathbf{W}(\mathbf{z}_{ri} + \mathbf{v}_{ri}) + \boldsymbol{\varepsilon}_{ri}, r = 1, 2, \cdots, M, \quad (6)$$

where $\mathbf{z}_{ri}|y_{ri} = k \sim N(\mu_k, \Sigma_k)$, and $\mathbf{v}_{ri}|\mathbf{v}_{[n_r]/i} \sim N(\mu_{v_{ri}}, m_{ri}^{-1}\Psi_r)$.

## Recovery of comparable gene expression matrices

Once the estimated cluster labels are obtained by PRECAST, we can remove unwanted variations using a set of housekeeping genes as negative control genes that are not affected by other biological effects[74]. In this study, we used a set of mouse/human housekeeping genes from the Housekeeping and Reference Transcript Atlas[75]. First, we obtained vectors, $\tilde{\mathbf{x}}_{ri} = (\tilde{\mathbf{x}}_{ri1}, \cdots, \tilde{\mathbf{x}}_{riL})$, of the expression of $L$ housekeeping genes by matching the set of housekeeping genes and genes passing QC for each dataset. By performing PCA, we obtained the top 10 PCs, $\widehat{\mathbf{h}}_{ri}$, as covariates to adjust for unwanted variation. One of the outputs of PRECAST, the posterior probability of $y_{ri}$ ($\widehat{\mathbf{r}}_{ri} \in \mathbb{R}^K$), can be used as the design matrix to explain biological variation between cell/domain types. Finally, we used a linear model for the

normalized gene expression vector

$$\mathbf{x}_{ri} = \boldsymbol{\alpha}\widehat{\mathbf{r}}_{ri} + \boldsymbol{\gamma}\widehat{\mathbf{h}}_{ri} + \boldsymbol{\varepsilon}_{ri}, \quad (7)$$

where $\boldsymbol{\alpha}$ is a $p$-by-$K$ dimensional matrix for biological effects between cell/domain types and $\boldsymbol{\gamma}$ is a $p$-by-10 dimensional matrix of regression coefficients associated with the unwanted factors. After obtaining the parameter estimates in Eqn. (7), users can remove batch effects from the original normalized gene expression using

$$\widehat{\mathbf{x}}_{ri} = \mathbf{x}_{ri} - \widehat{\boldsymbol{\gamma}}\widehat{\mathbf{h}}_{ri}.$$

This strategy can also be applied to samples from multiple biological conditions when such information is available. This can be achieved by adding additional covariates for biological conditions in Eq. (7).

## Differential expression analysis and enrichment analysis

After removing unwanted variation of gene expression matrices for multiple slides, we performed DE analyses and enrichment analysis on all four datasets. In detail, we used the FindAllMarkers function in the R package *Seurat* with default settings to detect the differentially expressed genes for each domain detected by PRECAST. The DE analysis was considered to identify domain-specific DE genes with adjusted $p$-values of less than 0.001 and a log-fold change of greater than 0.25. After obtaining a set of genes specific to each domain, we performed gene set enrichment analysis (GSEA) on a set of detected DE genes using g:Profiler with the g:SCS multiple testing correction method and applying a significance threshold of 0.05[76].

## Conditional SVG analysis

After obtaining aligned embeddings and domain labels for DLPFC and HCC Visium data using PRECAST, we detected the SVGs by adjusting the estimated aligned embeddings as covariates to investigate the role of SVGs beyond differences between cell/domain types. In detail, we used the function spark.vc in the R package *SPARK* to identify SVGs adjusted for cell/domain-relevant covariates for each sample. Finally, an FDR of 1% was adopted to identify the significant SVGs.

## Trajectory inference/RNA velocity analysis

To further investigate the development and differentiation of these identified spatial domains/cells by PRECAST, we used the aligned embeddings and domain clusters estimated by PRECAST to perform trajectory inference, or RNA velocity analysis if splicing and unsplicing information was available. For the DLPFC Visium data, mouse liver ST data and mouse OB Slide-seqV2 data, we conducted trajectory inference for the combined spots in all multiple tissue sections using Slingshot[77]. We inputted the aligned embeddings and domain clusters estimated by PRECAST into the function slingshot in the R package *slingshot* for implementation. Because the splicing and unsplicing information was available for the HCC Visium dataset, we ran RNA velocity analysis using the scvelo.tl.velocity function in the Python module *scvelo* based on the splicing and unsplicing matrices, then the domain clusters estimated by PRECAST were used to visualize the inferred RNA velocity and latent time.

## Cell-type deconvolution analysis

We performed deconvolution analysis using Robust Cell Type Decomposition (RCTD)[55], a supervised learning method used to decompose each spatial transcriptomics pixel into a mixture of individual cell types while accounting for platform effects. We leveraged the results of deconvolution analysis for better biological interpretation of the real data analysis.

For mouse liver ST data, we used scRNA-seq data on liver tissue sections from an adult mouse from the MCA[46]. By removing cell types

with cell numbers of less than 18, the reference data retained 4640 cells belonging to 17 cell types: B cells with high *Jchain* expression ($n = 43$), erythroid cells with high *Hbb-bs* expression ($n = 555$), peri-portal hepatocytes ($n = 26$), hepatocytes with high *Fabp1* expression ($n = 149$), Kupffer cells ($n = 1046$), erythroid cells with high *Hbb-bt* expression ($n = 62$), endothelial cells ($n = 1196$), granulocytes ($n = 194$), NK cells ($n = 114$), macrophages ($n = 191$), dendritic cells ($n = 422$), B cells with high *Fcmr* expression ($n = 97$), T cells ($n = 219$), plasmacytoid dendritic cells ($n = 90$), hepatocytes with high *Spp1* expression ($n = 99$), pericentral hepatocytes ($n = 119$), and hepatic stellate cells ($n = 18$).

For the mouse OB Slide-seqV2 data, we used 10X Genomics Chromium scRNA-seq data collected from the wild-type OB[52]. The reference data included 17,453 cells belonging to 40 cell types: three astrocyte types (Astro1: $n = 1087$; Astro2: $n = 129$; and Astro3: $n = 599$), two endothelial types (EC1: $n = 897$ and EC2: $n = 445$), two mesenchy-mal types (Mes1: $n = 200$ and Mes2: $n = 65$), three microglials (MicroG1: $n = 255$; MicroG2: $n = 766$; and MicroG3: $n = 294$), monocytes ($n = 238$), two murals (Mural1: $n = 43$ and Mural2: $n = 272$), myelinating-oligodendrocyte cells ($n = 294$), macrophages ($n = 197$), five olfactory ensheathing cell types (OEC1: $n = 812$; OEC2: $n = 664$; OEC3: $n = 1073$; OEC4: $n = 1083$; and OEC5: $n = 281$), oligodendrocyte progenitor cells (OPC: $n = 137$), red blood cells (RBCs: $n = 103$), and 18 neural subtypes (OSNs: $n = 467$; PGC-1: $n = 437$; PGC-2: $n = 146$; PGC-3: $n = 106$; GC-1: $n = 540$; GC-2: $n = 1245$; GC-3: $n = 96$; GC-4: $n = 80$; GC-5: $n = 516$; GC-6: $n = 89$; GC-7: 293; Immature: $n = 1150$; Transition: $n = 733$; Astrocyte-Like: $n = 1078$; M/TC-1: $n = 24$; M/TC-2: $n = 44$; M/TC-3: $n = 411$; and EPL-IN: $n = 64$). For more details of the cell types, please refer to Tepe et al.[52].

For the HCC Visium data, we leveraged a droplet-based scRNA dataset collected from HCC and intra-hepatic cholangiocarcinoma patients to serve as the reference data for deconvolution[61]. After fil-tering out 92 cells of unclassified cell type, the reference data con-tained 5023 cells belonging to seven cell types: B cells ($n = 598$), CAFs ($n = 724$), HPC-like cells ($n = 254$), malignant cells (702), T cells ($n = 1429$), TAMs ($n = 437$), and TECs ($n = 879$). After acquiring the deconvolution results from the above-mentioned reference, we com-bined the proportions of B cells, T cells, and TECs into a single set and referred to this as the immune cell proportion.

## Comparisons of methods

We conducted comprehensive simulation and real data analyses to compare PRECAST with existing methods of data integration and clustering.

We applied the following single-cell integration methods to benchmark the data integration performance of PRECAST: (1) Seurat V3[26]; (2) Harmony[23] implemented in the R package *harmony*; (3) fastMNN[24] implemented in the R package *batchelor*; (4) Scanorama[25] implemented in the Python module *scanorama*. (5) scGen[28]; (6) scVI[27] implemented in the Python module *scvi*; (7) MEFISTO[29] implemented in the Python module *mofax*; and (8) PASTE[30] implemented in the Python module *paste*. In the real data analyses, PRECAST and all other methods used the same list of selected genes from the preprocessing steps as input. The first six methods were designed for scRNA-seq integration, while MEFISTO can be used for scRNA-seq and SRT data integration, and PASTE is designed for integrating SRT data from multiple adjacent tissue slices into a single slice (see Supplementary Materials).

To evaluate clustering performance, we considered the following four methods using the extracted aligned embeddings as input: (1) SC-MEB implemented in the R package *SC.MEB*[20], (2) Louvain imple-mented in the R package *igraph*[78], (3) BayesSpace implemented in the R package *BayesSpace*[79], and (4) BASS implemented in the R package *BASS*[80]. SC-MEB and BayesSpace were recently developed to perform spatial clustering based on a discrete Markov random field[20,79], and

Louvain is a conventional non-spatial clustering algorithm based on community detection in large networks[78], while BASS was a newly developed clustering method for multiple SRT data based on the aligned embeddings from Harmony. In the implementation, we set the default values for them in the respective packages (see Supplementary Materials).

## Evaluation metrics

We evaluated the methods' performances in data integration, obtain-ing embeddings for cellular biological effects, and spatial clustering using the following metrics.

**Local inverse Simpson's index.** To assess performance in batch-effect removal, we used the cell-type/integration local inverse Simpson's index (LISI), cLISI, and iLISI[23] to quantify the performance in merging the shared cell populations among tissue slides and mixing spots from $M$ tissue slides. cLISI assigns a diversity score to each spot that repre-sents the effective number of cell types in the neighborhoods of that spot. For $M$ datasets with a total of $K$ cell types, accurate integration should maintain a cLISI value close to 1, reflecting the purity of the unique cell types in the neighborhood of each spot, as defined by the aligned embeddings. Erroneous embedding includes neighborhoods with a cLISI of more than 1, while the worst cases have cLISI close to $K$, suggesting that neighbors have $K$ different types of cells. iLISI has a similar form and implication as cLISI but is based on a sample index set rather than clusters of cell types. Thus, a larger iLISI value means the different samples have more sufficient mixing.

**Silhouette coefficient.** To simultaneously evaluate the separation of each cell/domain cluster and mixing of multiple datasets, we calcu-lated the average silhouette coefficient of the SRT datasets using two different groupings: (1) grouping using known cell types as the cell/domain-type silhouette coefficient ($silh_{cluster}$) and (2) grouping using different datasets as the batch silhouette coefficient ($silh_{batch}$). In the data integration, a larger value of $silh_{cluster}$ indicates better preserva-tion of the biological signals between cell/domain types, while a smaller value of $silh_{batch}$ suggests better mixing of datasets. These two metrics can be summarized using the F1 score as follows[81]:

$$\text{F1 score} = \frac{2(1 - silh'_{batch})silh'_{cluster}}{silh'_{cluster} + (1 - silh'_{batch})} \in [0,1], \tag{8}$$

where $silh'_{batch} = \frac{1 + silh_{batch}}{2}$ and $silh'_{cluster} = \frac{1 + silh_{cluster}}{2}$. A larger F1 score suggests better data integration that preserves the biological varia-tions between cell/domain types while removing other non-cellular biological variations across multiple tissues.

**Canonical correlation coefficients and/or conditional correlation.** For dimension reduction, we applied two measurements to assess the performance of true latent feature recovery in the simulation studies. The first was the mean canonical correlation between the estimated features and the true one, defined as

$$\text{CCor} = \frac{1}{q} \sum_{l=1}^{q} \zeta_l(\mathbf{z}_i, \hat{\mathbf{z}}_i), \tag{9}$$

where $\zeta_l$ is the $l$-th canonical correlation coefficient. The second mea-sure was the mean conditional correlation between gene expression $\mathbf{x}_i$ and cell type label $y_i$ given the estimated latent features $\hat{z}_i$, defined as

$$\text{ConCor} = \frac{1}{p} \sum_{j=1}^{p} corr(y_i, resid_{ij}), \tag{10}$$

where $resid_i$ is the residual of $x_{ij}$ regressing on $\hat{z}_i$, and $corr(y_i, resid_{ij})$ is the Pearson correlation coefficient between $y_i$ and $resid_{ij}$.

**Adjusted Rand index and/or normalized mutual information.** To evaluate performance in spatial clustering, we used ARI[82] and NMI[83]. ARI is the corrected version of the Rand index (RI)[84] and is used avoid some of the drawbacks of RI[82]. ARI measures the similarity between two different partitions and ranges from −1 to 1. A larger value of ARI means a higher degree of similarity between two partitions. ARI takes a value of 1 when the two partitions are equal up to a permutation. Whereas NMI is a variant of the mutual information (MI) that normalizes the value of MI to within the range of 0 and 1. When the two partitions are equal up to a permutation, NMI takes a value of 1.

## Simulations

*Scenario 1. Raw gene expression count data with different scales of batch effects: domain labels and spatial coordinates from Potts models.* For this scenario, we generated the raw gene expression count data for three samples, as well as the spatial coordinates based on Potts models with four neighborhoods. All quantities, such as the labels $y_{ri}$s and latent features $\boldsymbol{v}_{ri}$s, of spots were randomly simulated from the generative model (5). We generated the class label $y_{ri}$ for each $r = 1, 2, 3$ and $i = 1, \cdots, n_r$, with $n_r \in \{65 \times 65, 60 \times 60, 60 \times 60\}$, corresponding to rectangular lattices from a $K$-state ($K = 7$) Potts model with the smoothing parameter $\beta_r = 0.8 + 0.2(r - 1)$, using the function `sampler.mrf` from the R package *GiRaF*. Then, we generated latent features $\boldsymbol{v}_{ri}$s from the CAR model with the same $N_{ri}$ as defined in the Potts model and covariance matrix $\Psi'_r = (\sigma_{r,ij})$, with $\sigma_{r,ij} = r(0.2r)^{|i-j|}$, using the function `rmatrixnorm` in R package *LaplacesDemon*. We set different values for the smoothing parameter $\beta_r$ and covariance matrices $\Psi'_r$ across $r$, to mimic the heterogeneity of three samples. The domain labels $y_{ri}$s and latent features $\boldsymbol{v}_{ri}$s were fixed once they were obtained.

Following this, we generated domain-relevant latent features $\mathbf{z}_{ri}$ in the model (5) from the conditional Gaussian distribution, such that $\mathbf{z}_{ri}|y_{ri} = k \sim (\mu_k, \Sigma_k)$, where $\mathbf{z}_{ri} \in R^q$ with $q = 10$. Structures for $\mu_k$ and $\Sigma_k$ are shown in Supplementary Data 8. Next, we generated $\widetilde{\mathbf{W}} = (\widetilde{w}_{ij}, i \leq p, j \leq q)$ with each $\widetilde{w}_{ij} \overset{i.i.d.}{\sim} N(0,1)$, performed QR decomposition on $\widetilde{\mathbf{W}}$ such that $\widetilde{\mathbf{W}} = \widetilde{Q}\widetilde{R}$, and assigned $\mathbf{W} = \widetilde{Q}$, which is a column orthogonal matrix. Then, we generated a batch-loading matrix $\mathbf{W}_r$ by generating $\overline{\mathbf{W}}_r = \overline{\mathbf{W}} + \overline{E}_r$ with $\overline{\mathbf{W}} = (\overline{w}_{ij}, i \leq p = 2000, j \leq q_r), q_r = 2$, $\overline{w}_{ij} \overset{i.i.d.}{\sim} N(0,1)$, $\overline{E}_r = (\overline{e}_{rij}, i \leq p, j \leq q_r)$, and $\overline{e}_{rij} \sim N(0, \overline{\sigma}_r^2)$ with $\overline{\sigma}_1 = 0.5$, $\overline{\sigma}_2 = 0.8$, $\overline{\sigma}_3 = 1$. In a similar manner to the generation of $\mathbf{W}$, we performed orthogonalization of $\overline{\mathbf{W}}_r$ to generate $\mathbf{W}_r$. Next, we generated $\boldsymbol{\zeta}_{ri}$ in the model (5) by $\boldsymbol{\zeta}_{ri} = (\zeta_{ri1}, \cdots, \zeta_{riq_r})^\top$ such $\zeta_{rik} \overset{i.i.d.}{\sim} N(0, b_{scale}^2 \sigma_r^2)$ with $\sigma_1 = 1$, $\sigma_2 = 2$ and $\sigma_3 = 0.5$, where $b_{scale}$ controlled the scales of batch effects. Here, we considered three scales of batch effects, corresponding to low, middle and high, by taking the value of 1, 2, or 3 for $b_{scale}$, respectively.

Next, we generated a high-dimensional normalized expression matrix using $\mathbf{x}_{ri} = \tau_r + \mathbf{W}(\mathbf{z}_{ri} + \boldsymbol{v}_{ri}) + \mathbf{W}_r \boldsymbol{\zeta}_{ri} + \boldsymbol{\varepsilon}_{ri}, \tau_{rj} \sim N(0, 4), \boldsymbol{\varepsilon}_{ri} \sim N(\mathbf{0}, \Lambda_r)$, where $\tau_{rj}$ is the $j$-th element of $\tau_r$, $\Lambda_r = \text{diag}(\lambda_{rj})$, $j = 1, \ldots, p$, $\lambda_{1j} = 2(1 + 1.5|z_{1j}|)$ with $z_{1j} \overset{i.i.d.}{\sim} N(0,3)$; $\lambda_{2j} = 2(1 + z_{2j})$ with $z_{2j} \overset{i.i.d.}{\sim} U[0,1]$; and $\lambda_{3j} = 2(1 + 2z_{3j})$ with $z_{3j} \overset{i.i.d.}{\sim} U[0,1]$. Finally, we generated raw gene expressions $\tilde{x}_{ri} = (\tilde{x}_{ri1}, \cdots, \tilde{x}_{rip})^\top$ using $\tilde{x}_{rij} \sim Poisson(\exp(x_{rij}))$. The term $\boldsymbol{\varepsilon}_{ri}$ makes the distribution of $\tilde{x}_{rij}$ over-dispersed, which better imitates the properties of count expression. In this scenario, for each sample $r$, we only observed the raw expression $\tilde{x}_{rij}$ of gene $j$ and spot $i$ and spatial coordinates $s_{ri}$ for spot $i$.

*Scenario 2. Raw gene expression count data with different scales of batch effects: domain labels and spatial coordinates from DLPFC data.* To validate the generalizability of PRECAST, we also generated data based on three DLFPC datasets (ID: 151507, 151669, and 151673) from three donors (Visium platform). The domain labels $y_{ri}$s for spots were obtained from annotations made by Maynard et al.[21], together with the spatial coordinates. To generate the latent features $\boldsymbol{v}_{ri}$s, PRECAST was used to fit the three datasets, and we used the estimated features $\hat{\boldsymbol{v}}_{ri}$s as input. The cluster labels $y_{ri}$s and latent features $\boldsymbol{v}_{ri}$s were fixed once they were obtained. The other quantities were

generated in the same way as in scenario 1, but with a different $\mu_k$ (Supplementary Data 8).

*Scenario 3. Raw gene expression data: count matrix, domain labels and spatial coordinates from DLPFC data.* To make PASTE comparable to the other methods, we obtained raw gene expression data, domain labels and spatial coordinates from all three tissue slides (ID: 151673, 151674, and 151675) from the last donor. The domain labels were obtained from Maynard et al.[21]. As the true shared embeddings were unknown, we did not evaluate the canonical correlations. To generate the count matrix, we randomly added a pseudocount for each raw count from a binomial distribution with size three and probability parameter 0.3, similar to Zeira et al.[30]. Finally, we obtained the count matrix and spatial coordinates as input for the compared methods.

## Gene selection for integrative analysis

By performing QC, we filtered out genes with zero expression in multiple spots, and spots with zero expression of many genes (see "Data resources"). In our analyses, we used SPARK[40] to select top SVGs for human DLPFC Visium data, mouse liver ST data and HCC Visium data. However, we used SPARK-X[41] to select SVGs for mouse olfactory bulb Slide-seqV2 data, since SPARK cannot handle datasets with a large number of spots. In total, we selected the top 2000 SVGs for each sample using SPARK or SPARK-X. Next, we prioritized genes based on the number of times they were selected as SVGs in all samples and chose the top 2000 genes as input for PRECAST and the other compared analytical methods.

We used human DLPFC Visium data and mouse liver ST data, which had manual annotations, to confirm that spatially-aware gene selection methods did not represent a crucial part of PRECAST. Here, we selected the top 2000 highly variable genes (HVGs) for each sample using *FindVariableFeatures* with default settings in the *Seurat* R package. In addition, to examine the impact of different SVG selection methods to choose SVGs on the performance of PRECAST, we applied four methods to select the top 2000 SVGs for each sample of these two datasets. These methods included SPARK, SPARK-X, SpatialDE[42], and nnSVG[43].

## Data resources

**Human dorsolateral prefrontal cortex Visium data.** We downloaded spatial transcriptomic data for human DLPFC obtained on the 10x Visium platform from https://doi.org/10.5281/zenodo.4730634. These data were collected from 12 human postmortem DLPFC tissue sections from three independent neurotypical adult donors, and the raw expression count matrix contained 33,538 genes for each sample, with a total of 47,681 spatial locations. Before conducting the analysis, we first performed QC on each sample to filter out genes with non-zero expression levels for fewer than 20 spots and the spots with non-zero expression levels for less than 20 genes. The filtering step led to sets of 14,535 genes on average in a total of 47,680 spatial locations. The annotated spatial domains in all 12 samples based on the cytoarchitecture in the original study[21] were layer 1 ($n = 5321$), layer 2 ($n = 2858$), layer 3 ($n = 17,587$), layer 4 ($n = 3547$), layer 5 ($n = 7300$), layer 6 ($n = 6201$), white matter ($n = 4514$), and undetermined spots ($n = 352$). In the analysis, we treated these manual annotations as the ground truth to evaluate the clustering and data integration performance of the different methods.

**Mouse liver ST data.** We downloaded eight sets of mouse liver ST data from https://zenodo.org/record/4399655. The eight datasets contained 15,302 genes, on average, measured over 4865 spatial spots in total. In the QC steps, we first filtered out genes with non-zero expression levels in fewer than 20 spots and spots with non-zero expression levels for fewer than 20 genes. The filtering step led to a set of 9221 genes, on average, from a total of 4865 locations. The annotated spatial domains in all eight samples, based on marker genes in

the original study[44] were portal veins ($n = 1223$), central veins ($n = 720$), haemoglobin ($n = 464$), immune-related domain ($n = 163$), mesenchymal-related domain ($n = 110$), and undetermined spots ($n = 2183$), which were used to evaluate the clustering and data integration performances of the different methods in the analyses.

**Mouse olfactory bulb Slide-seqV2 data.** We obtained the data for 20 mouse OB Slide-seqV2 sections in Replicate 2 from CNBI accession number GSE169021 https://www.ncbi.nlm.nih.gov/geo/query/acc.cgi?acc=GSE169021. We used the first 16 slides because of the low quality of the last four slides. The data contained 21,571 genes, on average, over all 693,863 spatial locations across 16 sections, which were equally distributed along the anterior-posterior axis of the same mouse. In the QC steps, we first filtered out genes with non-zero expression levels in fewer than 20 spots and spots with non-zero expression levels for fewer than 20 genes. The filtering step led to a set of 14,307 genes, on average, for a total of 594,890 locations. To evaluate the impact of spatial resolution, we further collapsed nearby spots in each tissue slide using square grids of size 70 × 70. Then, the same QC steps were performed to obtain normalized expression for the analysis. To examine the structure of the mouse OB, we relied on the structural annotation in the Allen Brain Atlas[48] and used the results of downstream analyses to determine the specific regions of OB.

**Human hepatocellular carcinoma Visium data.** These data were from two tissue sections each from tumor and tumor-adjacent regions of an HCC patient, and contained 36,601 genes from over 9813 spatial locations. In the QC steps, we first filtered out genes with non-zero expression levels in fewer than 20 spots and spots with non-zero expression levels for fewer than 20 genes. The filtering step led to a set of 14,851 genes on average from a total of 9813 locations. In this data, manual annotations for the TNE and stroma regions were provided by a pathologist using the Visium companion H&E images. We further performed spatial deconvolution to examine the spatial distribution of malignant cells using RCTD[55].

**Reporting summary**
Further information on research design is available in the Nature Portfolio Reporting Summary linked to this article.

## Data availability
All datasets used in this study are publicly available. These include the 12 human dorsolateral prefrontal cortex Visium datasets (https://doi.org/10.5281/zenodo.4730634), eight mouse liver ST datasets (https://zenodo.org/record/4399655), 16 mouse OB Slide-seqV2 datasets (https://www.ncbi.nlm.nih.gov/geo/query/acc.cgi?acc=GSE169021) and four human hepatocellular carcinoma Visium datasets (Raw FASTQ data are available at https://www.ncbi.nlm.nih.gov/sra?linkname=bioproject_sra_all&from_uid=858545, and H&E images are available at https://doi.org/10.6084/m9.figshare.21280569.v1 and https://doi.org/10.6084/m9.figshare.21061990.v1). The structural annotation of mouse olfactory bulb is available at Allen Brain Atlas (https://atlas.brain-map.org/). All other relevant data supporting the key findings of this study are available within the article and its Supplementary Information files or from the corresponding author upon reasonable request. Source data are provided with this paper.

## Code availability
The PRECAST methods were implemented in an open-source, publicly available R package[85] that is available at https://cran.r-project.org/package=PRECAST and https://github.com/feiyoung/PRECAST. Code for reproducing the analysis can be found at https://github.com/feiyoung/PRECAST_Analysis.

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

## Acknowledgements

We thank Dr. Juan Zhou for critical reading and feedback. This work was supported by University Development Fund (UDF01003033) from The Chinese University of Hong Kong, Shenzhen; AcRF Tier 2 grant (MOE-T2EP20220-0009) from the Ministry of Education, Singapore, and grant from the National Natural Science Foundation of China (11931014). The computational work for this article was partially performed using resources from the National Supercomputing Centre, Singapore (https://www.nscc.sg).

## Author contributions

J.L. initiated and designed the study, W.L. implemented the model and developed the software tool with assistance from Y.Y., W.L., X.L., and Z.L. performed the simulation studies and the benchmark evaluation; J.L. wrote the manuscript, and W.L., X.L., Z.L., Y.Y., M.L., Y.J., X.S., W.Z., H.J., J.Y., and J.L. edited and revised the manuscript.

## Competing interests

The authors declare no competing interests.
