## [Peer Review File · Nature Communications]

REVIEWER COMMENTS

Reviewer #1 (Remarks to the Author):

In this Liu et al manuscript, the authors introduce their method PRECAST to perform data integration on multiple spatial transcriptomics (ST) datasets which can benefit downstream analysis including cell/domain detection and detection of spatially variable genes (SVGs). The idea is novel in the context of ST, but well grounded with the well-established benefits of harmonization methods in single cell RNA-sequencing literature. The manuscript is well written overall but some technical details need to be further clarified. I have the following specific comments.

1. It seems unclear to me how the ability of PRECAST to handle batch effects is evaluated in the simulation. The authors compare several scenarios with normalized/ raw gene expression and the source of the domain label which I believe is the input format for their software and maybe should be put in supplementary materials. I think the reader will be more interested in different scenarios where the batch effect has different scales/impact on the data integration since the main goal of PRECAST is to remove the batch effect.
2. For real datasets, how to guarantee that the heterogeneity across samples is purely due to technical effect but not true biological reasons? In the "Recovery of comparable gene expression matrices" part, the authors claim that additional covariates for biological conditions could be used in the model in order to remove batch effects. Which kind of biological condition is used in the analysis?
3. One main goal of PRECAST is to perform spatial clustering/ spatial domain detection. The authors only compare PRECAST with several methods designed for single cells and PASTE which was designed originally to focus on alignment (which is not feasible in PRECAST). The baseline performance for clustering methods using multi-sample is the performance of methods using single slides independently. The authors at least need to compare their performance with methods designed for ST data including BayesSpace, SpaGCN. In addition, there is already method design for multi-sample cell type clustering and spatial domain detection. Please check "Li, Z., Zhou, X. BASS: multi-scale and multi-sample analysis enables accurate cell type clustering and spatial domain detection in spatial transcriptomic studies. *Genome Biol* 23, 168 (2022). <https://doi.org/10.1186/s13059-022-02734-7>". It would be nice if the authors also compare the performance with BASS.
4. The authors showcase several downstream analyses that could be accomplished by their aligned embeddings. How is the performance compared to the performance when using the original gene expression/ embedding only using one slide? For example, the authors use SPARK to identify SVGs? What's the difference between using gene expression directly? How to quantify/guarantee the type-I error?

Reviewer #2 (Remarks to the Author):

Summary

Liu et al. present a new method, PRECAST, for joint integration and clustering of multiple-sample spatially-resolved transcriptomics (SRT) data. Unlike previous methods, the integration of multiple samples and the spatially-aware clustering are performed simultaneously within the same methodological framework. The authors demonstrate the performance of the method using a number of evaluation metrics on both simulated and previously published experimental data, and provide an R implementation from CRAN. The development of a method for integrated analysis of SRT datasets consisting of multiple samples is a particularly valuable contribution. However, some crucial aspects of the evaluations are currently not clear, which make it difficult to interpret the evaluation results and method performance.

Major comments

(1) In the evaluations using the four experimental datasets, it is not clear if the same preprocessing techniques have been used for the different methods, which makes it difficult to understand if this is a fair comparison. Specifically, the section “Data resources” in Methods explains that the methods SPARK (Sun et al. 2020) and SPARK-X (Zhu et al. 2021) have been used for preprocessing to select the top 2000 spatially variable genes (SVGs) for each dataset. These top 2000 SVGs are then used as the input for PRECAST. However, for all competing methods, it appears that non-spatial preprocessing techniques have been used instead (i.e. following the standard preprocessing tutorials for each method, which select the top highly variable genes (HVGs) instead of SVGs, where HVGs do not take into account any spatial information), although this is not completely clear from the description (“Comparisons of methods” in Methods). If so, then this is a biased and unfair comparison, which clearly favors PRECAST, since PRECAST is using spatially-aware preprocessing while the competing methods are using non-spatially-aware preprocessing. In effect, the superior performance of PRECAST could be due to the choice of preprocessing (SVGs instead of HVGs) instead of the method itself, or possibly a combination of the two. To clarify this, and/or to demonstrate the contribution of both the choice of preprocessing and the method itself, it would be useful to add a comparison either using HVGs (instead of SVGs) as preprocessing for PRECAST, or SVGs as preprocessing for the competing methods.

(2) A related comment concerns the choice of SPARK or SPARK-X for the preprocessing (to select SVGs for PRECAST). SPARK is used for 3 of the datasets, and SPARK-X for 1 dataset. However, SPARK and SPARK-X are very different methods, which can give substantially different sets of SVGs, depending on the dataset (e.g. see the evaluations in Zhu et al. 2021, or Weber et al. 2022). Therefore, similar to point 1 above, it would be useful to understand how sensitive the results are to the choice of SPARK vs. SPARK-X for preprocessing. One way to demonstrate this could be to evaluate performance using alternative methods for preprocessing to identify SVGs, such as SpatialDE (Svensson et al. 2018) or nnSVG (Weber et al. 2022). Alternatively, the same method (i.e. SPARK) could be used for all four datasets (instead of mixing SPARK and SPARK-X), although in this case the slow runtime of SPARK for larger datasets could be prohibitive.

(3) The section on “Code availability” mentions that code to reproduce the analyses is provided on GitHub. However, the linked repository does not appear to contain this code. (It contains a number of data objects and 4 individual code files, but does not contain code for most of the analyses.) Code to reproduce the analyses and figures should be added to this repository, along with a readme file to explain the contents.

(4) The section on “Data availability” mentions that all datasets are publicly available. However, one of the 4 experimental datasets is not available, and the simulated datasets are also not provided. These should be made publicly available.

(5) Supplementary Figure S2 provides some details on computational scalability (2000 genes, variable number of spots). However, this does not include the time for preprocessing to select SVGs with SPARK or SPARK-X. If preprocessing with SPARK is a crucial part of the method (i.e. if the superior performance of PRECAST strictly depends on preprocessing with SPARK, as discussed above in point 1), then some details should be provided on the additional computational time required to run these preprocessing steps. (In particular, SPARK scales cubically with the number of spots, which could mean prohibitive runtimes in practical terms for larger datasets. By contrast, SPARK-X, nnSVG, or non-spatial HVGs would all be much faster.) Similarly, it would be useful to know how runtime scales with the number of genes.

(6) In the trajectory inference / pseudotime evaluations, it would be useful to provide some more discussion on biological interpretation. For example, in the Visium DLPFC dataset, it is not clear that pseudotime-associated genes in mature cells within the cortex would necessarily be biologically meaningful, or whether these are spurious results. Some comments or discussion on the applicability

of the method for these analyses would be informative for readers.

Minor comments

(1) The online tutorials require some additional details to explain how users can load the datasets. (Currently, the tutorials include hard-coded paths to local files on the authors' computers, so the code cannot be run directly.)

(2) Top of page 9: the text mentions the number of cells per spot as 1-10 in Visium and 1-3 in Slide-seqV2. These numbers also depend on the tissue type and species, which could be clarified.

Reviewer #1:

1. In this Liu et al manuscript, the authors introduce their method PRECAST to perform data integration on multiple spatial transcriptomics (ST) datasets which can benefit downstream analysis including cell/domain detection and detection of spatially variable genes (SVGs). The idea is novel in the context of ST, but well grounded with the well-established benefits of harmonization methods in single cell RNA-sequencing literature. The manuscript is well written overall but some technical details need to be further clarified. I have the following specific comments.

Our response #1:

Thank you for your constructive comments. We have provided further clarification of these technical details in the following point-by-point response.

2. It seems unclear to me how the ability of PRECAST to handle batch effects is evaluated in the simulation. The authors compare several scenarios with normalized/ raw gene expression and the source of the domain label which I believe is the input format for their software and maybe should be put in supplementary materials. I think the reader will be more interested in different scenarios where the batch effect has different scales/impact on the data integration since the main goal of PRECAST is to remove the batch effect.

Our response #2:

This is a very good suggestion, and one we chose to follow. Considering that scVI and PASTE can only be applied to count expression, we varied the scales of batch effects in both scenarios 2 and 4. For ease, we reordered scenarios 2, 4 and 5 to 1, 2 and 3, respectively, and scenarios 1 and 3 to 4 and 5, respectively. For the ‘new’ scenarios 1 and 2, we considered three different scales of batch effects (low, middle and high) and made comparisons between PRECAST and other methods for expression count matrices with these various scales of batch effects. In detail, we generated batch effect term ζ_{ri} in the model

$$\mathbf{x}_{ri} = \tau_r + W(\mathbf{z}_{ri} + \nu_{ri}) + W_r \zeta_{ri} + \varepsilon_{ri},$$

with $\zeta_{ri} = (\zeta_{ri1}, \dots, \zeta_{riq_r})^\top$ such that $\zeta_{rik} \stackrel{i.i.d.}{\sim} N(0, b_{scale}^2 \sigma_r^2)$ and $\sigma_1 = 1, \sigma_2 = 2$ and $\sigma_3 = 0.5$. Here, b_{scale} controls the scales of batch effects, and we used values of 1, 2 or 3 for b_{scale} corresponding to the low, middle and high batch effects. Finally, we generated raw gene expression data $\tilde{\mathbf{x}}_{ri} = (\tilde{x}_{ri1}, \dots, \tilde{x}_{rip})^\top$ using $\tilde{x}_{rij} \sim \text{Poisson}(\exp(x_{rij}))$. The term ε_{ri} makes the distribution of $\tilde{\mathbf{x}}_{rij}$ over-dispersed, which can better imitate the properties of count expression. For full details concerning data generation, see Methods Section in the revised main text.

As shown in Fig. R1c, in scenarios 1 and 2, all methods became worse in terms of data integration (top panel) and clustering (bottom panel) when the scale of batch effects increased. PRECAST achieved the best performance in comparison with other methods, with the highest F1 score for the average silhouette coefficients and ARI. Moreover, estimation of the embeddings induced by neighboring microenvironments by PRECAST became worse as the scale of batch effects increased (middle panel). Fig. R2a & b show the data integration and extraction of embeddings for biological effects between cell/domain types in scenarios 1 and 2, respectively. PRECAST achieved better data integration for the combination of iLISI and cLISI and obtained better or comparable embeddings in terms of both canonical and conditional correlations compared with other methods. The canonical correlations (CCor) measure the association

between the estimated features and the underlying true ones, with a larger CCor being considered better, while the conditional correlations (ConCor) measure how much information on domain/cell clusters the estimated features contain, with a smaller ConCor being better. Fig. R3a shows the clustering performance in scenarios 1 and 2. We observed that the clustering performance for all methods decreased as the scale of batch effects increased, in terms of both ARI and NMI, but PRECAST outperformed other methods. In conclusion, the simulation results in scenarios 1 and 2 suggest that data integration, embedding extraction and clustering worsened for all methods as the scale of batch effects increased, but PRECAST consistently equalled or outperformed other methods in both scenarios.

Changes we made : We changed Fig. 1 in the main text to the Fig. R1 and changed Supplementary Fig. S1&2 to Fig. R2&3.

Figure R1. Schematic overview of PRECAST and simulation results. (a) PRECAST is a unified probabilistic factor model that simultaneously estimates aligned embeddings and cluster labels, with consideration of spatial smoothness in both the cluster label and low-dimensional embedding spaces. Normalized gene expression matrices from multiple tissue slides are used as input. (b) Representative PRECAST downstream analyses. (c) In the simulations, we investigated two ways to generate spatial coordinates and cell/domain labels for count matrices: Potts models (scenario 1) and three cortex tissues from the DLPFC data (scenario 2). We examined the impact of scales in batch effects (low, middle, and high) on the data integration performance using scenarios 1 and 2. We also considered an additional scenario 3, which was favorable for PASTE. We evaluated performance in terms of data integration, the estimation of aligned embeddings, and the estimation of slide-specific embeddings due to neighboring microenvironments and spatial clustering, using average silhouette coefficient (F1) scores and canonical correlation coefficients (CCor). The adjusted Rand index (ARI) and ARIs displayed for the other methods were evaluated based on the results of the spatial clustering method SC-MEB. PRECAST outperformed all other data integration methods in scenarios 1 and 2, and its performance was comparable to PASTE in scenario 3. In simulations, only PRECAST estimated the slide-specific embeddings. We also evaluated the CCor with underlying truth in scenarios 1 and 2.

Figure R2. Batch correction and dimension reduction performance for simulated data. (a) Violin plot of cLISI/iLISI based on the batch-corrected 15-dimensional embeddings from PRECAST and eight other compared methods in scenarios 1-3. (b) Bar plot of canonical/conditional correlations based on the batch-corrected 15-dimensional embeddings from PRECAST and eight other methods in scenarios 1-3. In scenario 3, the true low-dimensional embeddings were unknown, so we could not evaluate the canonical correlation (CCor). (c) Violin plot of cLISI/iLISI/F1 score based on the batch-corrected 15-dimensional embeddings from PRECAST and five other compared methods in scenarios 4 and 5. (d) Violin plot showing the canonical correlations between estimated slide-specific embeddings due to neighboring microenvironments from PRECAST, and the underlying truth. Bar plot showing canonical/conditional

correlations based on the batch-corrected 15-dimensional embeddings from PRECAST and five other methods in scenarios 4 and 5. scVI and PASTE are only applicable to scenarios 1-3 with count matrices.

Figure R3. Clustering analysis in simulated data. (a) Domain clustering performance of PRECAST and eight other integration methods, based on SC-MEB clustering, in scenarios 1-3. Upper panel: Violin plot of ARIs from SC-MEB clustering, based on the low-dimensional embeddings of Harmony, fastMNN, Scanorama, scGen, scVI, MEFISTO and PASTE. Middle panel: Violin plot of NMIs from PRECAST and other compared methods. Bottom panel: Bar plot showing the number of clusters selected by PRECAST and other compared methods, where the true number of clusters is $K = 7$. (b) Domain clustering performance of PRECAST and eight other integration methods, based on SC-MEB clustering in scenarios 4 and 5. scVI and PASTE are only applicable to scenarios 1-3 with count matrices.

3. For real datasets, how to guarantee that the heterogeneity across samples is purely due to technical effect but not true biological reasons? In the “Recovery of comparable gene expression matrices” part, the authors claim that additional covariates for biological conditions could be used in the model in order to remove batch effects. Which kind of biological condition is used in the analysis?

Our response #3:

Thank you for raising this point. By demonstrating that non-cellular biological effects, such as batch effects and/or biological effects between conditions, can be projected into the cellular biological space, PRECAST implicitly accounts for these projections using an intrinsic CAR component. For each tissue slide, we assume that the normalized expressions in each sample (r) can be broken down into additional parts with respect to non-cellular biological effects, as follows:

$$\mathbf{x}_{ri} = W(\mathbf{z}_{ri} + \boldsymbol{\nu}_{ri}) + W_r \boldsymbol{\zeta}_{ri} + \boldsymbol{\varepsilon}_{ri}, \quad (1)$$

where $\boldsymbol{\nu}_{ri}$ is a q -dimensional vector that captures the spatial dependence among neighboring spots; $W_r \in \mathbb{R}^{p \times \tilde{q}}$ is a loading matrix for a factor related to non-cellular biological effects; and $\boldsymbol{\zeta}_{ri}$, independent of $(\mathbf{z}_{ri}, \boldsymbol{\nu}_{ri})$, is the corresponding \tilde{q} -dimensional vector. If cell biological space (W) and non-cellular biological space (W_r) is orthogonal, $W_r \boldsymbol{\zeta}_{ri}$ can be included in the random errors $\boldsymbol{\varepsilon}_{ri}$. On the other hand, if cell biological space and non-cellular biological space are non-orthogonal, we can project W_r s onto the column spaces of W , i.e., $\widehat{W}_r = W(W^T W)^{-1} W^T W_r$, and the approximated model can be derived as

$$\mathbf{x}_{ri} \approx W(\mathbf{z}_{ri} + \mathbf{v}_{ri}) + \boldsymbol{\varepsilon}_{ri}. \quad (2)$$

Technical details of this approximation can be found in the Methods Section.

Since PRECAST estimates aligned low-dimensional embeddings by removing non-cellular biological effects such as batch effects, and simultaneously performs spatial clustering, its output includes aligned embeddings and cluster labels for each spot in each sample. To further perform differential expression analysis for all combined slides, we need to remove batch effects for each individual gene. Following (Risso et al., 2014), we obtained the top 10 principal components (PCs) from housekeeping genes. By taking the top 10 PCs and the posterior probability of y_{ri} ($\widehat{\mathbf{r}}_{ri} \in \mathbb{R}^K$) from PRECAST as covariates, we are able to remove unwanted variations using:

$$\mathbf{x}_{ri} = \widehat{\mathbf{r}}_{ri}^T \boldsymbol{\alpha} + \widehat{\mathbf{h}}_{ri}^T \boldsymbol{\gamma} + \boldsymbol{\varepsilon}_{ri}. \quad (3)$$

After obtaining the parameter estimates in Eqn. (3), users can remove batch effects from the original normalized gene expression using

$$\widehat{\mathbf{x}}_{ri} = \mathbf{x}_{ri} - \widehat{\mathbf{h}}_{ri}^T \widehat{\boldsymbol{\gamma}}.$$

Note that in Risso et al. (2014), this relies on the availability of cell-type labels if users want to keep variations in cell biological effects. Here, if additional covariates for biological conditions are available, one may perform linear regression analysis to remove the biological effects on conditions in a similar fashion as mentioned above. In our analysis, there are no biological conditions that exist across multiple slides. Thus, we do not remove such effects in the analysis.

4. One main goal of PRECAST is to perform spatial clustering/ spatial domain detection. The authors only compare PRECAST with several methods designed for single cells and PASTE which was

designed originally to focus on alignment (which is not feasible in PRECAST). The baseline performance for clustering methods using multi-sample is the performance of methods using single slides independently. The authors at least need to compare their performance with methods designed for ST data including BayesSpace, SpaGCN. In addition, there is already method design for multi-sample cell type clustering and spatial domain detection. Please check “Li, Z., Zhou, X. BASS: multi-scale and multi-sample analysis enables accurate cell type clustering and spatial domain detection in spatial transcriptomic studies. *Genome Biol* 23, 168 (2022). <https://doi.org/10.1186/s13059-022-02734-7>”. It would be nice if the authors also compare the performance with BASS.

Our response #4:

This is a good question. We thank the reviewer for raising the point, and we have added more results to make comprehensive comparisons. The reviewer is correct that we primarily compare PRECAST with data integration methods designed for scRNA-seq. In contrast to other data integration methods in scRNA-seq/SRT, PASTE aims to align adjacent tissues from the same individual or to combine small pieces of each slide into a single slide, but cannot be applied to integrate tissue slides from different individuals. PASTE needs the spatial coordinates of the array to be fixed and assumes that the expression matrix is low rank, which does not account for the randomness of gene expression. PRECAST is better at aligning embeddings underlying shared cell/domain clusters by removing effects that are not relevant to cluster types, such batch effects, while making spatial transcriptomics of multiple tissue slides comparable, even if these come from different individuals. To some extent, the aim of PRECAST is similar to many other data integration methods used in scRNA-seq, such as Harmony, fastMNN, and Seurat V3, with the addition of a unique feature to estimate slide-specific embeddings that capture spatial dependence in neighboring cells/spots.

Thank you for highlighting the recent literature on BASS. We omitted comparison with it as BASS was published after our initial submission. In detail, BASS relies on Harmony to remove batch effects on the top PCs from expression levels, and applies spatial clustering on the aligned embeddings from Harmony. Thus, it can only be considered a spatial clustering method with consideration of both spatial domains and cell types. Since BASS is based on Harmony, to perform a fair comparison, we evaluated its clustering performance in comparison with other (spatial) clustering methods, including SC-MEB, BayesSpace, and Louvain. We based this on the embeddings from Harmony and PRECAST, respectively, in the analysis of both simulation and real data.

One drawback of BASS is that it cannot automatically select the number of clusters, so we used the number of clusters selected by PRECAST during the implementation of BASS to enable comparison. In the simulations, we observed that the clustering methods based on embeddings from PRECAST had larger ARI and NMI values than those based on embeddings from Harmony (Fig. R4a), suggesting that embeddings from PRECAST carried more information than embeddings from Harmony. Fig. R5 and R6 compare the use of embeddings from PRECAST and Harmony, respectively, for clustering in 12 human DLPFC sections and eight mouse liver sections. We also found that embeddings from PRECAST was more informative than those from Harmony, and PRECAST outperformed BASS based on embeddings from Harmony in clustering analysis.

Following your suggestions, we performed additional simulation studies for the single-sample clustering analysis using SC-MEB, BayesSpace, and SpaGCN, under scenarios 1-5, as shown in the following Fig.

R7. We found that PRECAST, as a joint clustering method, generally performed better than single-sample methods (Fig. R7). Furthermore, one of the biggest drawbacks of single-sample clustering analysis is that the identified clusters are not matched for all samples, meaning that it is necessary to perform manual assignment to match between-sample clusters. This can be labor-intensive, but PRECAST can automatically achieve this.

Changes we made : We added Fig. R4 to the Supplementary Figures, as Fig. S3. We added Figs. R5 and R6 to the Supplementary Figures as Fig. S9, and S24, respectively. We also added discussion concerning BASS in the main text (pages 5, 7, 8 and 19).

Figure R4. : Clustering analysis based on embeddings from PRECAST and Harmony in scenario 4, and scalability analysis. (a) Left panel: Box plot of ARIs from PRECAST, and BASS, SC-MEB, BayesSpace and Louvain based on the low-dimensional embeddings of PRECAST and Harmony. Middle panel: Box plot of NMIs from these methods. Right panel: Bar plot of the number of clusters selected by PRECAST, SC-MEB, BayesSpace and Louvain. BASS cannot choose the number of clusters automatically, so we used the number of clusters selected by PRECAST. (b) Linear computational complexity of PRECAST with regard to the number of spots/genes. Left panel: Line plot of running time and number of spots (given 2000 genes) when running 30 iterations of three datasets on a linux server with 2.10GHz Intel(R) Xeon(R) Gold 6230 CPU and 50G memory. Right panel: Line plot of running time and number of genes (given 15,000 spots in total) when running 30 iterations of three datasets on the same machine.

Figure R5. Clustering analysis based on different embeddings for the 12 dorsolateral prefrontal cortex Visium sections. Upper panel: Box plot of ARIs/NMIs for each sample from PRECAST, BASS, SC-MEB, BayesSpace and Louvain clustering, based on the low-dimensional embeddings of PRECAST and Harmony. Bottom panel: Bar plot of ARI/NMI for combined samples from BASS, SC-MEB, BayesSpace and Louvain clustering, based on the low-dimensional embeddings of PRECAST and Harmony.

Figure R6. Clustering analysis for the eight mouse liver sections, based on different embeddings. Upper panel: Box plot of ARIs/NMIs for each sample from PRECAST, BASS, SC-MEB, BayesSpace and Louvain clustering, based on the low-dimensional embeddings of PRECAST and Harmony. Bottom panel: Bar plot of ARI/NMI for combined samples from BASS, SC-MEB, BayesSpace and Louvain clustering, based on the low-dimensional embeddings of PRECAST and Harmony.

Figure R7. Comparison of clustering performance of PRECAST and other single-sample clustering meth-

ods (SC-MEB, BayesSpace and SpaGCN) in the simulations for 'new' scenarios 1-5 (scenarios 1-2 only presented the case with low batch effects because of similar patterns for two other cases).

5. The authors showcase several downstream analyses that could be accomplished by their aligned embeddings. How is the performance compared to the performance when using the original gene expression/ embedding only using one slide? For example, the authors use SPARK to identify SVGs? What's the difference between using gene expression directly? How to quantify/guarantee the type-I error?

Our response #5:

Thank you for these suggestion. In our study, the aligned embeddings were used in three downstream tasks: visualization, conditional spatial variation analysis (SVA) and trajectory inference. We evaluated the performance of these three tasks using embeddings from either PCA or DR-SC for each slide and aligned embeddings from PRECAST, to show the advantage of joint modelling by PRECAST. PCA is a widely used method to extract embeddings (Yang et al., 2022; Zhao et al., 2021) while DR-SC (Liu et al., 2022) is a newly developed method to extract embeddings from a single slide by performing joint dimension reduction and spatial clustering for SRT data.

To visualize the two-dimensional embeddings across multiple slides, we plotted the tSNE plots of embeddings from PRECAST in comparison with those from PCA and DR-SC in all four datasets, as shown in Fig. R8 and R9. We observed that visualization of embeddings by PRECAST was much better than that achieved by PCA and DR-SC for each slide, with domain clusters well-segregated and samples well-mixed.

To identify spatially variable genes (SVGs) that could not be explained by domain differences, we also performed conditional spatial variation analysis (SVA) in both LIBD human DLPFC data and HCC data generated using 10x Visium. For the purpose of comparison, we identified SVGs by either (1) using only gene expression, or (2) using gene expression with adjustment for embeddings from PRECAST. SPARK software reports the adjusted p -values to control false discovery rate (FDR) using the Benjamini–Yekutieli procedure. In our empirical studies, we selected SVGs using an adjusted p -value cutoff of 0.01. Since embeddings from PRECAST only carry information for cell/domain clusters, SVA adjusting for embeddings from PRECAST can identify SVGs with nonlaminar patterns. SVGs obtained by (2) almost represented a subset of SVGs obtained by (1), as shown in Fig. R10. Furthermore, we compared the QQ-plots in SVG testing from (1) and (2). We observed that nominal p -values using SPARK in both (1) and (2) were inflated. Inflation was less severe in SVA with adjustment for embeddings from PRECAST (2) than in SVA without adjustment (1), as shown in Fig. R11. By performing enrichment analysis for genes identified with adjustment for embeddings by PRECAST (2), we found that the SVGs detected in all 12 LIBD slices were highly enriched in many common KEGG pathways, such as Parkinson disease, Alzheimer disease, Huntington disease and multiple neurodegenerative diseases (Supplementary Fig. S15), cerebellum/cerebral cortex-related HPA pathways (Fig. S16), and cytoplasmic transition, cytosolic ribosome and structure molecular activity GO pathways (Fig. S17-S19), This suggests that the embeddings for biological effects between cell/domain types were effectively aligned by PRECAST.

To demonstrate that the aligned embeddings from PRECAST facilitated trajectory inference, we showed that pseudotime inferred using the aligned embeddings was "sample-invariant" in LIBD human DLPFC

data (Fig. R12). For each spatial domain identified, we observed similar pseudotime distributions across all 12 tissue sections using the aligned PRECAST embeddings, while pseudotime distributions substantially varied in trajectory inference for single slides across all 12 tissue sections when using embeddings from either PCA or DR-SC.

Changes we made : We added Fig. R12 to the Supplementary Figures, numbered S14. The main text we changed was on page 7.

Figure R8. Scatter plot of two-dimensional tSNEs of embeddings from PCA, DR-SC and PRECAST for each sample. (a) the 12 DLPFC Visium sections; and (b) the eight mouse liver sections.

Figure R9. Scatter plot of two-dimensional tSNEs of embeddings from PCA, DR-SC and PRECAST for each sample. (a) the 16 mouse olfactory bulb sections with reduced resolution; and (b) the four Human HCC sections.

Figure R10. Comparison of conditional SVA analysis and unconditional SVA analysis. Let A and B be the sets of conditional SVGs and unconditional SVGs, respectively. 12 DLPFC Visium sections: (a) Left panel: bar plot of the ratio of $\#A \cap B$ to $\#A$ for each sample; Right panel: bar plot of the ratio of $\#A \cap B$ to $\#B$ for each sample. (b) Venn diagram of two sets A and B for each sample. Four Human HCC sections: (c) Left panel: bar plot of the ratio of $\#A \cap B$ to $\#A$ for each sample; Right panel: bar plot of the ratio of $\#A \cap B$ to $\#B$ for each sample. (d) Venn diagram of two sets A and B for each sample.

Figure R11. Comparison of QQplot in conditional SVA testing and unconditional SVA testing. (a) 12 human DLPFC Visium sections. The first row was for slides from first donor with sample ID 151507-151510; the second row second donor and the last row last donor. (b) Four human HCC sections.

Figure R12. Comparison of the inferred pseudotime with embeddings from PRECAST, DR-SC and PCA for the 12 dorsolateral prefrontal cortex Visium sections. PCA and DR-SC were performed on each slide to obtain the embeddings, while PRECAST was performed on all slides to obtained aligned embeddings

Reviewer #2:

Major comments:

1. In the evaluations using the four experimental datasets, it is not clear if the same preprocessing techniques have been used for the different methods, which makes it difficult to understand if this is a fair comparison. Specifically, the section “Data resources” in Methods explains that the methods SPARK (Sun et al. 2020) and SPARK-X (Zhu et al. 2021) have been used for preprocessing to select the top 2000 spatially variable genes (SVGs) for each dataset. These top 2000 SVGs are then used as the input for PRECAST. However, for all competing methods, it appears that non-spatial preprocessing techniques have been used instead (i.e. following the standard preprocessing tutorials for each method, which select the top highly variable genes (HVGs) instead of SVGs, where HVGs do not take into account any spatial information), although this is not completely clear from the description (“Comparisons of methods” in Methods). If so, then this is a biased and unfair comparison, which clearly favors PRECAST, since PRECAST is using spatially-aware preprocessing while the competing methods are using non-spatially-aware preprocessing. In effect, the superior performance of PRECAST could be due to the choice of preprocessing (SVGs instead of HVGs) instead of the method itself, or possibly a combination of the two. To clarify this, and/or to demonstrate the contribution of both the choice of preprocessing and the method itself, it would be useful to add a comparison either using HVGs (instead of SVGs) as preprocessing for PRECAST, or SVGs as preprocessing for the competing methods.

Our response #1:

This is a very good question. It is essential to make “apples to apples” comparisons. In fact, we applied the same preprocessing procedures to obtain input gene expressions for all methods. In short, the input for PRECAST and the other methods was the same. Details about the preprocessing steps are given in the Methods Section on page 21. We apologize for this confusion.

Next, we evaluated the impact of using HVG to select genes. Using the two manually annotated datasets (human DLPFC Visium data and mouse liver ST data), we added comparison between PRECAST and other methods by taking the top 2,000 HVGs as the input. We obtained the top 2,000 HVGs for each sample using *FindVariableFeatures* with default settings in the *Seurat* R package. Then, we prioritized genes based on the number of times they were selected as HVGs across all samples, and chose the top 2,000 genes as the input for all methods. We observed that PRECAST consistently outperformed other methods (Fig. R13 and R14). Specifically, for the human DLPFC Visium data, Fig. R13a shows that the estimated embeddings from PRECAST carried more information about the domain labels (smaller ConCor). Fig. R13b shows that PRECAST achieved the best F1 score (a trade-off between cLISI and iLISI). As shown in Fig. R13c, PRECAST also achieved the highest ARI and NMI for both individual samples and combined samples. For the mouse liver ST data, Fig. R14a shows that the estimated embeddings from PRECAST carried comparable information about the domain labels; Fig R15b shows that PRECAST achieved the best F1 score (a trade-off between cLISI and iLISI), and Fig. R14c shows that PRECAST achieved the highest ARI and NMI for both individual samples and combined samples. In conclusion, PRECAST consistently outperformed other methods when using the top 2,000 HVGs.

We also evaluated the impact of using different methods to select the top genes to use as input for PRE-

CAST. PRECAST was robust when using different methods to select top genes for input (Figs. R16-17). More details for this can be found in our response #2.

Changes we made : We have revised the preprocessing steps in the Methods Section. We added Figs. R13-14 to the Supplementary Figures, numbered S20 and S29, respectively. The main text we changed was on pages 7, 9, 18, and 21.

Figure R13. Dimension reduction with batch correction and clustering analysis using top 2000 HVGs for the 12 dorsolateral prefrontal cortex Visium sections. (a) Boxplot/violin plot of conditional correlations from PRECAST and eight other methods. (b) Boxplot of F1 score (F1 score of the average silhouette coefficients), cLISI and iLISI for PRECAST and eight other methods. (c) Boxplot/violin plot of ARIs/NMIs for each sample from PRECAST and SC-MEB clustering, based on the low-dimensional embeddings of other compared methods, and bar plot of ARIs/NMIs for 12 combined samples from PRECAST and other methods.

Figure R14. Dimension reduction with batch correction and clustering analysis using the top 2,000 HVGs for the eight mouse liver sections. (a) Boxplot/violin plot of conditional correlations for PRECAST and eight other methods. A lower conditional correlation score is better. (b) Boxplots of F1 score (higher is better), cLISI (lower is better) and iLISI (higher is better) for PRECAST and eight other methods. (c) Boxplot/violin plot of ARIs/NMIs (higher is better) for each sample from PRECAST and SC-MEB clustering based on the low-dimensional embeddings of other compared methods, and bar plot of ARIs/NMIs for 12 combined samples for PRECAST and other methods.

2. A related comment concerns the choice of SPARK or SPARK-X for the preprocessing (to select SVGs for PRECAST). SPARK is used for 3 of the datasets, and SPARK-X for 1 dataset. However, SPARK and SPARK-X are very different methods, which can give substantially different sets of SVGs, depending on the dataset (e.g. see the evaluations in Zhu et al. 2021, or Weber et al. 2022). Therefore, similar to point 1 above, it would be useful to understand how sensitive the results are to the choice of SPARK vs. SPARK-X for preprocessing. One way to demonstrate this could be to evaluate performance using alternative methods for preprocessing to identify SVGs, such as SpatialDE (Svensson et al. 2018) or nnSVG (Weber et al. 2022). Alternatively, the same method (i.e. SPARK) could be used for all four datasets (instead of mixing SPARK and SPARK-X), although in this case the slow runtime of SPARK for larger datasets could be prohibitive.

Our response #2:

Thank you for these constructive suggestions. Using the two manually annotated datasets (human DLPFC Visium data and mouse liver ST data), we examined the impact of SVG selection methods on PRECAST.

We applied the four suggested methods, SPARK, SPARK-X, SpatialDE, and nnSVG, to select the top 2,000 SVGs for each sample. Then, we prioritized genes based on the number of times they were selected as SVGs across all samples, and chose the top 2,000 genes as the input for PRECAST. Then, we compared the performance of PRECAST based on SVGs from different methods. As shown in Fig. R15a & R16a, nnSVG resulted in the longest running time and spent ~0.64 hours on average on DLPFC samples (Fig. R15a). This was consistent with the time spent by nnSVG (~0.70 hour) for one DLPFC sample in the original paper; see Supplementary Figure S14 in Weber et al. (2022). Most importantly, PRECAST was robust to the preprocessing steps in the selection of SVGs, as shown in Fig. R15 & R16. Specifically, for human DLPFC Visium data, Fig. R16b, 16c and 16d show that the estimated embeddings carried comparable information about the domain labels and that data integration performance (i.e., F1 score, cLISI and iL- ISI) was comparable; as was clustering performance for both individual samples and combined samples (i.e., ARI and NMI). Similar results for mouse liver ST data are presented in Fig. R16b, 16c and 16d. In all, Taken together, these results suggest that PRECAST was robust to differences in preprocessing steps when selecting the top genes for input.

Changes we made : We added discussion about this in the main text. We added Figs. R16-17 to the Supplementary Figures, numbered S21 and S30, respectively. The main text we altered was on pages 8, 9 and 21.

Figure R15. Performance comparison of results from PRECAST using SPARK, SPARK-X, SpatialDE, nnSVG and HVG gene selection methods, for the 12 dorsolateral prefrontal cortex Visium sections. (a) Barplot of running times for each sample (left panel) and all samples. (b) Boxplot/violin plot of conditional cor-

relations. (c) Boxplot of F1 score of the average silhouette coefficients, cLISI and iLISI. (d) Boxplot/violin plot of ARIs/NMIs for each sample, and bar plot of ARIs/NMIs for 12 combined samples.

Figure R16. Comparison of PRECAST performance using five gene selection methods: SPARK, SPARK-X, SpatialDE, nnSVG and HVGs, to analyze the eight mouse liver sections. (a) Barplot of running times for each sample (left panel) and all samples. (b) Boxplot/violin plot of conditional correlations. (c) Boxplot of F1 score, cLISI and iLISI. (d) Boxplot/violin plot of ARIs/NMIs for each sample, and bar plot of ARIs/NMIs for the eight combined samples.

3. The section on “Code availability” mentions that code to reproduce the analyses is provided on GitHub. However, the linked repository does not appear to contain this code. (It contains a number of data objects and 4 individual code files, but does not contain code for most of the analyses.) Code to reproduce the analyses and figures should be added to this repository, along with a readme file to explain the contents.

Our response #3:

Thank you for your suggestions. We have uploaded the code to enable reproducibility of the analyses, and have included three folders named Simulation (simulated examples), Real_data_analysis (real data analysis) and Real_data_results (real data results visualization). Furthermore, we also provided a readme file to explain the contents. Please see the repository at https://github.com/feiyong/PRECAST_A_analysis.

4. The section on “Data availability” mentions that all datasets are publicly available. However, one of the 4 experimental datasets is not available, and the simulated datasets are also not provided. These should be made publicly available.

Our response #4:

Thank you for your suggestion. We have uploaded the hepatocellular carcinoma Visium data at https://www.ncbi.nlm.nih.gov/sra?linkname=bioproject_sra_all&from_uid=858545 (Raw FASTQ data), <https://doi.org/10.6084/m9.figshare.21280569.v1> and <https://doi.org/10.6084/m9.figshare.21061990.v1> (H&E images), and the simulated datasets in the Simulation folder of https://github.com/feiyoungh/PRECAS_T_Analysis.

Changes we made : We added the link to the four hepatocellular carcinoma Visium datasets to the 'Data availability' section in the main text.

5. Supplementary Figure S2 provides some details on computational scalability (2000 genes, variable number of spots). However, this does not include the time for preprocessing to select SVGs with SPARK or SPARK-X. If preprocessing with SPARK is a crucial part of the method (i.e. if the superior performance of PRECAST strictly depends on preprocessing with SPARK, as discussed above in point 1), then some details should be provided on the additional computational time required to run these preprocessing steps. (In particular, SPARK scales cubically with the number of spots, which could mean prohibitive runtimes in practical terms for larger datasets. By contrast, SPARK-X, nnSVG, or non-spatial HVGs would all be much faster.) Similarly, it would be useful to know how runtime scales with the number of genes.

Our response #5:

Thank you for these suggestions. We have tested the performance of PRECAST with different gene selection methods, such as HVGs, SPARK, SPARK-X, SpatialDE and nnSVG, in human dorsolateral prefrontal cortex Visium data and mouse liver ST data. PRECAST was robust to differences in preprocessing when selecting genes. Thus, preprocessing with SPARK is not a crucial part of the PRECAST method. We also determined the time taken for preprocessing to select genes, with HVGs and SPARK-X being the fastest (Fig. R16a & R17b). Thus, we recommend these two methods for preprocessing to select genes.

In addition, we demonstrated that PRECAST was of linear computational complexity to the number of genes, and it only took ~2.5 minutes to analyze a dataset with 2,000 genes and 15,000 spots for a fixed number of clusters (Fig. R17b, right panel).

Changes we made : We added Figure R17 to the Supplementary Figures, numbered S3.

Figure R17. Clustering analysis based on embeddings from PRECAST and Harmony in scenario 4, and scalability analysis. (a) Left panel: Box plot of ARIs from PRECAST, and BASS, SC-MEB, BayesSpace and Louvain based on the low-dimensional embeddings of PRECAST and Harmony. Middle panel: Box plot of NMIs from these methods. Right panel: Bar plot of the number of clusters selected by PRECAST, SC-MEB, BayesSpace and Louvain. BASS cannot choose the number of clusters automatically, so we used the number of clusters selected by PRECAST. (b) Linear computational complexity of PRECAST with regard to the number of spots/genes. Left panel: Line plot of running time and number of spots (given 2000 genes) when running 30 iterations of three datasets on a linux server with 2.10GHz Intel(R) Xeon(R) Gold 6230 CPU and 50G memory. Right panel: Line plot of running time and number of genes (given 15,000 spots in total) when running 30 iterations of three datasets on the same machine.

6. In the trajectory inference / pseudotime evaluations, it would be useful to provide some more discussion on biological interpretation. For example, in the Visium DLPFC dataset, it is not clear that pseudotime-associated genes in mature cells within the cortex would necessarily be biologically meaningful, or whether these are spurious results. Some comments or discussion on the applicability of the method for these analyses would be informative for readers.

Our response #6:

Thank you for these suggestions. In the DLPFC Visium dataset, we identified 858 genes associated with the estimated pseudotime with adjusted p -values of <0.001 . We first conducted gene set enrichment analysis based on the GO database and found that the pseudotime-associated genes identified by PRECAST were significantly enriched in nervous system development. Next, we examined the roles of the top

10 genes ordered by the adjusted p -values and the absolute correlation values between gene expression and the inferred pseudotime. These genes included *Mobp*, *Gfap*, *Mag*, *Mbp*, *Plp1*. *Mobp* encodes myelin-associated oligodendrocyte basic protein, and its expression increases in the first years of life (Primiani et al., 2014). *Gfap* encodes glial fibrillary acidic protein and plays an important role in the developing human brain (Mamber et al., 2012). *Mag* encodes myelin-associated glycoprotein, which is involved in myelin maintenance and glia-axon interactions, and serves an important role in the adult central nervous system (Lossos et al., 2015). A relative gradient of *Mbp* transcription is found within the developing human brain, from caudal to rostral (Kamholz et al., 1988). Myelin proteolipid protein gene (*Plp1*) expression is temporally regulated in the brain, and peaks during the active myelination period during central nervous system development (Pereira et al., 2013).

Changes we made : We added discussion about this point in the main text.

Minor comments:

7. The online tutorials require some additional details to explain how users can load the datasets. (Currently, the tutorials include hard-coded paths to local files on the authors' computers, so the code cannot be run directly.)

Our response #7:

Thank you for your suggestions. In the online tutorials, we have added details and provided a convenient way to access the data. All codes in the tutorials, including loading datasets, can now be run directly: see <https://feiyoung.github.io/PRECAST/> for more details.

8. Top of page 9: the text mentions the number of cells per spot as 1-10 in Visium and 1-3 in Slide-seqV2. These numbers also depend on the tissue type and species, which could be clarified.

Our response #8:

Thank you for your comments. Following your suggestions, we have revised these statements: see Page 9.

References

- J. Kamholz, J. Toffenetti, and R. Lazzarini. Organization and expression of the human myelin basic protein gene. *Journal of neuroscience research*, 21(1):62–70, 1988.
- W. Liu, X. Liao, Y. Yang, H. Lin, J. Yeong, X. Zhou, X. Shi, and J. Liu. Joint dimension reduction and clustering analysis of single-cell RNA-seq and spatial transcriptomics data. *Nucleic Acids Research*, 03 2022. ISSN 0305-1048. doi: 10.1093/nar/gkac219. URL <https://doi.org/10.1093/nar/gkac219>.
- A. Lossos, N. Elazar, I. Lerer, O. Schueler-Furman, Y. Fellig, B. Glick, B.-E. Zimmerman, H. Azulay, S. Dotan, S. Goldberg, et al. Myelin-associated glycoprotein gene mutation causes pelizaeus-merzbacher disease-like disorder. *Brain*, 138(9):2521–2536, 2015.
- C. Mamber, W. Kamphuis, N. L. Haring, N. Peprah, J. Middeldorp, and E. M. Hol. *Gfap* δ expression in glia of the developmental and adolescent mouse brain. *PLoS One*, 7(12):e52659, 2012.

- G. B. Pereira, F. Meng, N. T. Kockara, B. Yang, and P. A. Wight. Targeted deletion of the antisilencer/enhancer (ase) element from intron 1 of the myelin proteolipid protein gene (plp1) in mouse reveals that the element is dispensable for plp1 expression in brain during development and remyelination. *Journal of neurochemistry*, 124(4):454–465, 2013.
- C. T. Primiani, V. H. Ryan, J. S. Rao, M. C. Cam, K. Ahn, H. R. Modi, and S. I. Rapoport. Coordinated gene expression of neuroinflammatory and cell signaling markers in dorsolateral prefrontal cortex during human brain development and aging. *PLoS One*, 9(10):e110972, 2014.
- D. Risso, J. Ngai, T. P. Speed, and S. Dudoit. Normalization of rna-seq data using factor analysis of control genes or samples. *Nature Biotechnology*, 32(9):896–902, 2014.
- L. M. Weber, A. Saha, A. Datta, K. D. Hansen, and S. C. Hicks. nnsvg: scalable identification of spatially variable genes using nearest-neighbor gaussian processes. *bioRxiv*, 2022.
- Y. Yang, X. Shi, W. Liu, Q. Zhou, M. Chan Lau, J. Chun Tatt Lim, L. Sun, C. C. Y. Ng, J. Yeong, and J. Liu. Sc-meb: spatial clustering with hidden markov random field using empirical bayes. *Briefings in bioinformatics*, 23(1):bbab466, 2022.
- E. Zhao, M. R. Stone, X. Ren, J. Guenthoer, K. S. Smythe, T. Pulliam, S. R. Williams, C. R. Uyttingco, S. E. Taylor, P. Nghiem, et al. Spatial transcriptomics at subspot resolution with bayesspace. *Nature Biotechnology*, pages 1–10, 2021.

REVIEWERS' COMMENTS

Reviewer #1 (Remarks to the Author):

The authors have carefully addressed my comments. I have no additional comments. The authors should be applauded for this valuable piece of work!

Reviewer #2 (Remarks to the Author):

The authors have addressed all comments raised in the previous round of review.

Additional minor comment: gene names should be written in italics and upper case (for human genes), and italics with first letter capitalized and subsequent letters lower case (for mouse genes). This convention has not been followed in some locations (e.g. lines 255-256, 266, 273).

Reviewer #1:

1. The authors have carefully addressed my comments. I have no additional comments. The authors should be applauded for this valuable piece of work!

Our response #1:

Thank you very much for your encouraging words.

Reviewer #2:

1. Additional minor comment: gene names should be written in italics and upper case (for human genes), and italics with first letter capitalized and subsequent letters lower case (for mouse genes). This convention has not been followed in some locations (e.g. lines 255-256, 266, 273).

Our response #1:

Thank your for your very useful suggestion. We have carefully checked full text and revised them in the new version.